# The desensitization pathway of GABA$_A$ receptors, one subunit at a time

Marc Gielen [1,2 ✉], Nathalie Barilone[1] & Pierre-Jean Corringer[1]

GABA$_A$ receptors mediate most inhibitory synaptic transmission in the brain of vertebrates. Following GABA binding and fast activation, these receptors undergo a slower desensitization, the conformational pathway of which remains largely elusive. To explore the mechanism of desensitization, we used concatemeric α1β2γ2 GABA$_A$ receptors to selectively introduce gain-of-desensitization mutations one subunit at a time. A library of twenty-six mutant combinations was generated and their bi-exponential macroscopic desensitization rates measured. Introducing mutations at the different subunits shows a strongly asymmetric pattern with a key contribution of the γ2 subunit, and combining mutations results in marked synergistic effects indicating a non-concerted mechanism. Kinetic modelling indeed suggests a pathway where subunits move independently, the desensitization of two subunits being required to occlude the pore. Our work thus hints towards a very diverse and labile conformational landscape during desensitization, with potential implications in physiology and pharmacology.

[1] Channel Receptors Unit, Institut Pasteur, CNRS UMR 3571, 25 rue du Docteur Roux, 75015 Paris, France. [2] Sorbonne Université, 21 rue de l'École de Médecine, 75006 Paris, France. ✉email: marc.gielen@pasteur.fr

GABA$_A$ receptors (GABA$_A$Rs) are the main inhibitory synaptic receptors in the forebrain of vertebrates, and are involved in key physiological and pathological processes such as memory, epilepsy, anxiety, and sedation. This is well illustrated by their medical significance, since the most prevalent GABA$_A$Rs are the target of the widely used benzodiazepine class of drugs[1].

GABA$_A$Rs belong to the pentameric ligand-gated ion channel (pLGIC) superfamily, which also comprises the anionic glycine receptor, as well as the excitatory 5HT$_3$ serotonin receptors and the nicotinic acetylcholine receptors (nAChRs)[2]. Upon agonist binding, their transmembrane pore quickly opens to enable the selective flow of permeant ions across the plasma membrane, thereby affecting cell excitability. However, during sustained binding of the agonist, most pLGICs will gradually enter a shut-state refractory to activation, called the desensitized state, thereby preventing excessive activation[3]. The exact roles of desensitization in vivo are still debated, but potentially include the reduction of responses during high-frequency neurotransmitter release[4], the prolongation of synaptic currents[5], as well as the modulation of extra-synaptic receptors subjected to tonic activation by low ambient concentrations of neurotransmitters[6].

Recent functional and structural studies, mostly performed on anionic pLGICs, provide compelling evidence for a "dual-gate" model, in which the transmembrane domain (TMD) of pLGICs contains both an activation-gate, located in the upper half of the channel, and a desensitization-gate, located at the intracellular end of the channel[3,7–11]. Structural work on homopentameric receptors always showed symmetrical structures[7,9,11], while the recent structures of the heteromeric GABA$_A$ receptor show important asymmetric features within the extracellular domain (ECD)[10], but still a strong pseudo-symmetrical organization of the TMD. The current view of the dual-gate model thus supports that resting, active, and desensitized states are essentially symmetrical at the level of the TMD, desensitization involving, in the lower part of the channel, a movement of all subunits to occlude the permeation pathway. However, desensitization is a multi-phasic process, since the sustained application of agonist elicits currents that desensitize with several distinct decay time constants, which are usually portrayed by the existence of "fast" and "slow" desensitized states (noted $D_{fast}$ and $D_{slow}$ below, respectively)[3,12–15]. The structural rearrangements underlying these distinct desensitization components remain elusive. In particular, it is currently unknown whether subunits rearrange in a concerted manner, with $D_{fast}$ and $D_{slow}$ reflecting distinct states at the single-subunit level, or whether individual subunits can rearrange independently with distinct time courses. The first scheme would predict that pLGICs only visit pseudo-symmetrical states during desensitization, while the latter would imply that desensitization involves asymmetrical states.

To examine the contribution of individual subunits, we herein introduced gain-of-desensitization mutations in each individual subunit, both one-by-one and in combinations, and assessed their interplay during desensitization. We selected mutations nearby the desensitization-gate, which were previously found to specifically alter the desensitization kinetics and amplitude, without significant alteration of the upstream activation process. Since stereotypical synaptic GABA$_A$Rs are composed of two α, two β, and one γ subunits[16,17], targeting a single α- or β-subunit within the pentamer is out of reach using classical site-directed mutagenesis approaches. To circumvent this problem, we used a concatemeric construct, whereby all five subunits are connected by polyglutamine linkers. Owing to the fixed organization of subunits within this concatemer, we could introduce and combine gain-of-desensitization mutations in a defined manner, ensuring the perfect homogeneity of the resulting recombinant GABA$_A$R

populations. We generated a library of 26 combinations of mutated subunits, recorded their macroscopic desensitization kinetics, and analyzed the data by Markov-chain kinetics simulations.

## Results

### A pentameric concatemer recapitulates the biphasic desensitization profile of the GABA$_A$R reconstituted from loose subunits. To force the subunit arrangement, we used a previously described[18] concatemer consisting of β2–α1–β2–α1–γ2 subunits fused together with 15- to 20-residues long polyglutamine linkers. When assembled in the counter-clockwise orientation as seen from the extracellular space, it shows a canonical organization with two GABA binding sites at the β2–α1 interfaces and one benzodiazepine site at the α1–γ2 interface (Fig. 1a). In contrast, in the clockwise orientation, the concatemer would carry a single GABA binding site and no benzodiazepine-binding site. This orientation, if it occurs, should therefore yield minimal, if any, GABA-gated currents and no benzodiazepine-potentiation. We previously showed that expression of the concatemer in oocytes yields robust GABA-elicited currents with an apparent affinity for GABA and a potentiation by benzodiazepines similar to that of GABA$_A$Rs expressed from loose subunits[18]. This shows that the counter-clockwise assembly largely dominates the electrophysiological response. This innocuity towards the pharmacology of extracellular ligands also suggests that the inter-subunit linkers leave the ECD conformational dynamics unaffected.

To record desensitization kinetics at the best possible temporal resolution using Two-electrode voltage clamp (TEVC) recordings of Xenopus laevis oocytes, we minimized the dead volume of our set-up and applied a supersaturating GABA concentration (10 mM), thereby optimizing the onset of electrophysiological responses in the 20–25 ms timescale (20–80% current rise times). As discussed in a previous publication, TEVC recordings of Xenopus laevis oocytes are well-suited to the study of desensitization of pLGICs owing to the robustness of the approach, which contrasts with the very high inter- and intracellular variability when using patch-clamp methods[3]. Recordings of the wild-type concatemer show robust currents, with desensitization profiles indistinguishable from that of conventional α1β2γ2 GABA$_A$Rs assembled from unconnected subunits (Fig. 1b, c; Supplementary Table 1; see ref. [8]), further arguing that the linkers do not affect the conformational changes at play during desensitization. Desensitization shows two well-separated components that are perfectly resolved by our procedure, a fast ($τ_{fast} = 4.8 ± 1.2$ s) and a slow one ($τ_{slow} = 24.4 ± 7.8$ s). The amplitude of the former carries about a third of the total desensitization amplitude, yielding a weighted desensitization time constant ($τ_w$) of about 18 s. After one minute of GABA application, the residual current accounted for about 10% of the peak current (Fig. 1b, c; Supplementary Table 1).

### Single desensitizing mutations show contrasting phenotypes depending on their location within the pentamer. For gain-of-desensitization mutations in α1, β2, and γ2 subunits, we chose the valine mutation at the 5′ position of the third transmembrane segment (M3), namely α1$^{N307V}$ on α1-subunits (SU2 and SU4), β2$^{N303V}$ on β2-subunits (SU1 and SU3), and γ2$^{H318V}$ on the single γ2-subunit (SU5) (Fig. 1d–f)—this prime notation, akin the one largely used for the M2 segment, starts at the cytoplasmic end of the M3 segment[19]. Indeed, we previously showed that these mutations markedly speed up the desensitization of α1β2γ2 GABA$_A$Rs[8]. We also showed that mutations in this region of the TMD do not alter significantly the concentration–response curve of the GABA-elicited peak currents, measured before the onset of

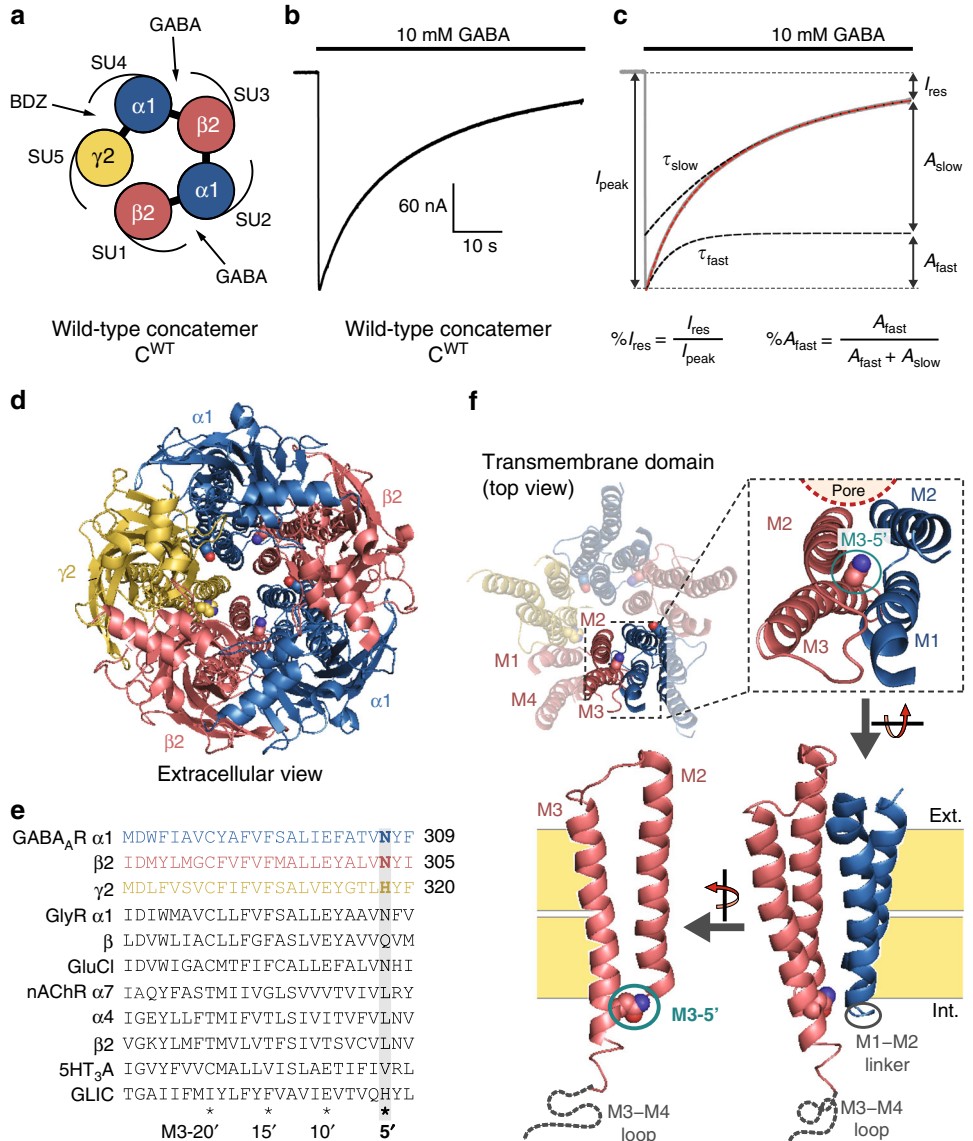

**Fig. 1 The wild-type α1β2γ2 pentameric GABA$_A$R concatemer. a** Schematic top view of the concatemer. The two β2/α1 ECD interfaces (SU1/SU2 and SU3/SU4) harbor the two GABA-binding sites, while the α1/γ2 ECD interface (SU4/SU5) contains the benzodiazepine-binding site. **b** Representative TEVC recording of a *Xenopus laevis* oocyte expressing the wild-type concatemer, C$^{WT}$. **c** Depiction of the experimental values used to quantify desensitization: $τ_{fast}$ and $τ_{slow}$ are the time constants of fast and slow desensitization components, respectively; %$A_{fast}$ is the relative amplitude of the fast component; %$I_{res}$ is the relative residual current after 1 min of 10 mM GABA application. Of note, the weighted desensitization time constant can be defined as $τ_w = \%A_{fast} * τ_{fast} + (1 - \%A_{fast}) * τ_{slow}$. **d** Cryo-EM structure of the α1β3γ2 GABA$_A$R (pdb 6I53[17]), as seen from the extracellular space. The β2 and β3 GABA$_A$ subunits are highly homologous, and both display an asparagine residue at the M3-5′ position. Note the central pore, lined by the M2 helices of the five subunits, forming the transmembrane channel. **e** Sequence alignment of the M3 segment of various pLGIC subunits. All sequences are the mouse orthologs, except GLIC (*Gloeobacter violaceus*), as well as the α4 and β2 nAChR subunits (human). The M3-5′ residues, mutated in the present study, are highlighted (gray box; bold characters for GABA$_A$ subunits). **f** Enlarged view of the α1β3γ2 GABA$_A$R structure highlighting the location of the M3-5′ residue at the M2/M3 transmembrane interface as seen from the side of the channel, facing the M1–M2 linker of the adjacent subunit.

desensitization. This indicates only a weak effect of the mutations on the resting-to-active state transition, and a major effect on the active-to-desensitized state transition.

Mutations were introduced one at a time on the concatemer. We define C$^{WT}$ as the wild-type concatemer, C$^i$ the concatemer with a single M3-5′ valine mutation on subunit number $i$, and C$^{ij}$ the concatemer where subunits $i$ and $j$ are both mutated, up to C$^{12345}$ where all subunits are mutated (Fig. 2a).

For the single mutations, C$^1$ (SU1 = β2) and C$^2$ (SU2 = α1) display desensitization kinetics similar to that of C$^{WT}$, while constructs C$^3$, C$^4$, and C$^5$ displayed robust gain-of-desensitization phenotypes (Fig. 2; Supplementary Fig. 1 and Supplementary

Table 1), yielding weighted desensitization rates of 6.2, 3.4, and 3.3 s, respectively, as compared to 18 s for C$^{WT}$. The three mutations accelerate fast desensitization by about 2-fold and slow desensitization by about 3-fold ($τ_{fast}$ = 2.7, 2.9, and 2.9 s; $τ_{slow}$ = 7.1, 7.3, and 7.2 s for C$^3$, C$^4$, and C$^5$, respectively). C$^4$ and C$^5$ in addition increase the relative amplitude of the fast component (% $A_{fast}$ = 20.0%, 86.7%, and 86.3% for C$^3$, C$^4$, and C$^5$, respectively), explaining their stronger effect. Of note, the C$^5$ construct displays an identical desensitization phenotype compared to the single mutant α1β2γ2$^{H318V}$ expressed from unconnected subunits[8], which is consistent with the assumption that our concatemeric design does not affect the desensitization properties of GABA$_A$Rs,

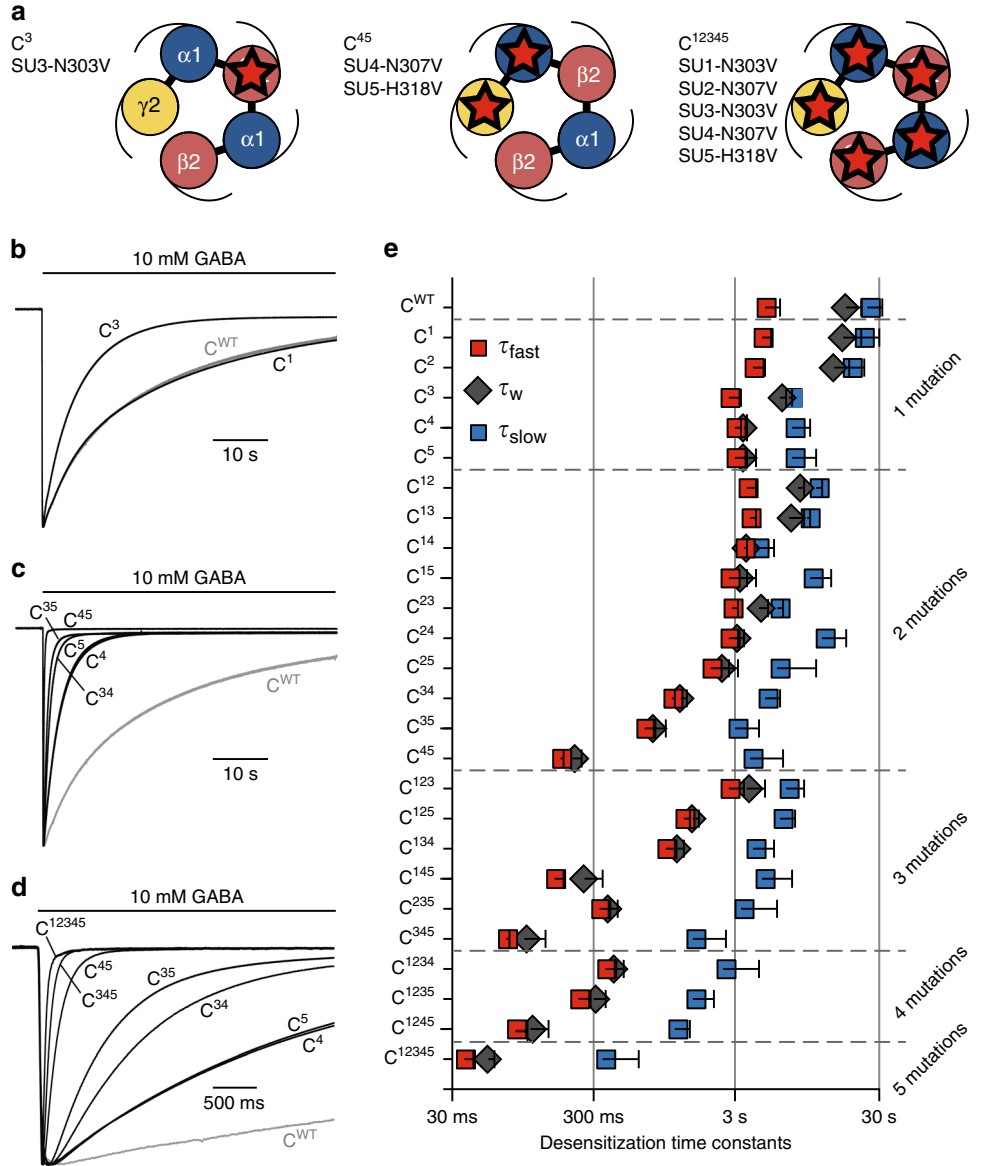

**Fig. 2 Desensitization kinetics of α1β2γ2 concatemers harboring combinations of M3-5′ valine mutations. a** Schematic top views of the C³ (left), C⁴⁵ (middle) and C¹²³⁴⁵ (right) concatemers. **b–d** Representative TEVC recording of *Xenopus laevis* oocytes expressing the indicated concatemers. Note the change in timescale for recordings in panel (**d**). **e** Plot indicating the mean values for fast (red squares), slow (blue squares), and weighted (dark gray diamonds) desensitization time constants for the indicated concatemers. Error bars are standard deviations. See Supplementary Fig. 1 for individual data points and Supplementary Table 1 for numerical values, the number of cells, and number of independent series of experiments.

even in the context of receptors harboring M3-5′ mutations. This is unsurprising, since the linkers are located in the extracellular part, and cannot interact directly with the M3-5′ residues located at the intracellular end of the pore.

It is noteworthy that the mutations are located at the cytoplasmic end of the TMD, with the side-chain of the mutated residue facing the M1–M2 linker of the neighboring subunit (Fig. 1f). Therefore, C¹, C², C³, C⁴ and C⁵ are mutated at β2-α1, α1-β2, β2-α1, α1-γ2, and γ2-β2 interfaces, respectively. The different mutations being introduced at different interfaces, it was expected that they display different phenotypes. However, the difference between C¹ and C³ is surprising, since they both correspond to mutations at the β2-α1 interface, showing virtually identical microenvironment. This indicates that the effect of the single mutations not only depends on the nature of the mutated interface, but also on the particular position of the mutated subunit within the pentamer.

**Combining mutated subunits increases desensitization kinetics and reveals synergistic effects.** To investigate the functional interaction between mutations at the various interfaces, we built an extensive library of twenty-six cDNAs including concatemers comprising two mutations (ten different constructs), three mutations (six constructs), four mutations (four constructs) or five mutations (one single construct, C¹²³⁴⁵), and assessed their desensitization profile as described above (Fig. 2; Supplementary Fig. 1 and Supplementary Table 1).

Recordings confirmed the modest effect of SU1 and SU2 mutations, which produce small effects when performed on concatemers with background mutations at other subunits (0.8 to 1.8-fold decrease in $\tau_w$ for SU1 and 1.1 to 2.8-fold for SU2, among 9 background-mutated concatemers for both). They also confirm the intermediate effect of SU3 (2 to 6.3-fold decrease in $\tau_w$ among 10 background-mutated concatemers), and the marked effect of SU4 and SU5 (effect of 5–16-fold among 9 and 10 mutated

concatemers for SU4 and SU5, respectively; Supplementary Fig. 2).

In all cases, combining gain-of-desensitization mutations together adds up to increase desensitization kinetics. For instance, the double mutant $C^{45}$ displays a fast desensitization component ($\tau_{\text{fast}} = 180$ ms) 26-fold faster than $C^{\text{WT}}$, accounting almost entirely for the overall desensitization (%$A_{\text{fast}} = 98.8\%$), and a barely measurable steady-state current (%$I_{\text{res}} = 0.8\%$). Such phenotype is further strengthened by mutating SU3: $C^{345}$ desensitizes with an even faster desensitization component in the 70 ms timescale. Mutating all five subunits gave a slightly more profound phenotype, with a fast desensitization component of 40 ms (see construct $C^{12345}$; Fig. 2d, e; Supplementary Table 1). Of note, for constructs akin $C^{345}$ and $C^{12345}$, the fast component is so fast that we probably miss a sizeable fraction of the peak current, thereby overestimating the amplitude of the slow desensitization component and the measurement of the relative steady-state current. Also, the steady-state current values and the amplitudes of the slow desensitization components are barely measurable for such constructs, rendering the related values (%$I_{\text{res}}$ and $\tau_{\text{slow}}$) unreliable.

To investigate the additivity of the various mutations' effects, we first compared the effect of individual mutations on the weighted desensitization kinetics of different concatemers with background mutations (Supplementary Fig. 2). While this analysis is crude, the series of double mutants already suggests some level of inter-subunit coupling. Indeed, while the SU1 mutation barely affects the desensitization of $C^{\text{WT}}$, it increases the weighted desensitization kinetics of $C^2$ by 75%, thereby hinting towards a coupling between SU1 and SU2. More strikingly, SU4 mutation speeds up desensitization about 5-fold on both $C^{\text{WT}}$, $C^1$, $C^2$, and $C^3$ backgrounds, while it increases the weighted desensitization kinetics of $C^5$ by 15-fold, clearly hinting towards synergistic effects of SU4 and SU5 mutations.

Second, we compared the desensitization profiles of $C^{34}$ and $C^{35}$. Since mutating SU4 or SU5 yields identical desensitization phenotypes (Fig. 2c–e; Supplementary Table 1), $C^{34}$ and $C^{35}$ should yield identical phenotypes if the effects of mutations were additive. Our data contradict such hypothesis, since both desensitization components of $C^{35}$ are faster than the ones of $C^{34}$, resulting in a 55% faster weighted desensitization rate (Fig. 2c–e; Supplementary Table 1). Thus, the effects of mutating

the M3-5′ residues are non-additive, especially for SU3 and SU4 or SU3 and SU5 subunit combinations.

**The conformational pathway of desensitization involves asymmetrical and non-concerted quaternary motions: implementation of a general model.** The present analysis unravels two key features governing the desensitization kinetics.

First, the markedly different effects observed upon mutation of SU1 and SU3, which both involve homologous mutations that are located in identical micro-environments, show that strongly asymmetrical motions are involved in the desensitization pathway. Since SU3 mutation has a strong effect on desensitization, the structural reorganization at this interface appears to be a limiting process. In contrast, mutation in SU1 has a very weak effect, suggesting either a small structural reorganization at this level, or, more likely, that the structural reorganization would not be rate limiting (see Discussion).

Second, the marked non-additive nature of the mutations, as discussed above, is not compatible with a concerted mechanism. Indeed, in such a scheme, the effect of mutations should directly translate their impact on the free energy landscape of the receptor, and should thus be additive.

As an illustration, we attempted to fit the whole set of data with a concerted model, in which the receptors can only visit a handful of pseudo-symmetrical conformations that include a fast and a slow desensitized state (Supplementary Fig. 3a–e). Here and throughout the manuscript, each model was built as a Markov-chain kinetic scheme and the whole-cell currents activated by a supersaturating concentration of GABA were simulated using the software QUB[20] (Supplementary Table 2). However, adjusting the parameters to correctly fit the desensitization of $C^{\text{WT}}$, $C^4$, and $C^5$, did not account for their synergistic effect since the simulated $C^{45}$ $\tau_{\text{fast}}$ and $\tau_{\text{w}}$ values are respectively 4.7 and 4.1-fold higher than the values observed experimentally (Supplementary Fig. 3f–g).

To implement the asymmetric and non-concerted properties, we turned to a radically different scheme in which all subunits can desensitize independently from the other subunits (Fig. 3). In this model, each subunit can enter its desensitized conformation while the other subunits are either in their open or desensitized conformations. For simplicity, we decided to implement only the desensitization of SU3, SU4, and SU5, since these subunits are by

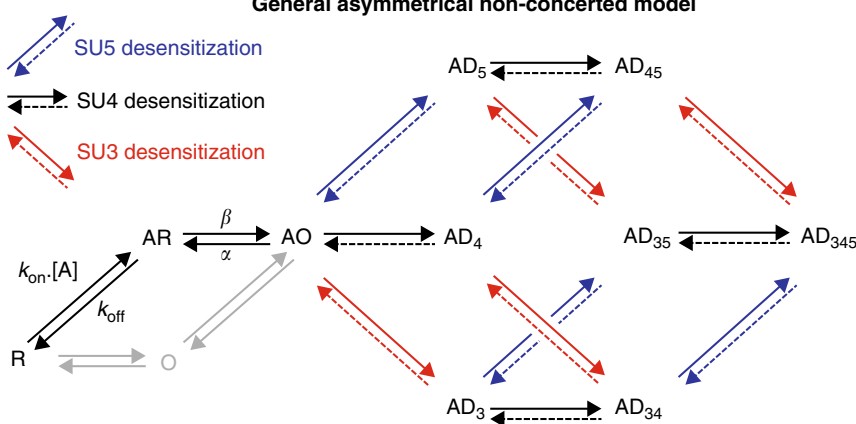

**Fig. 3 General scheme for the simulation of desensitization: an asymmetric non-concerted model.** The first part in the kinetic scheme is the binding of the agonist A to the resting state R, which favors the opening of the channel (AO state) with a gating efficacy $E = \beta/\alpha$. Of note, unliganded openings do exist but are not taken into account for our kinetic modeling as they barely contribute to the electrophysiological response (see main text). We also only include one binding event, even though $\alpha1\beta2\gamma2$ $\text{GABA}_A\text{Rs}$ contain two binding sites whose occupation is required for substantial activation. Upon channel opening, the receptor can then transit from a fully activated AO state to states where only one subunit enters its desensitized conformation ($\text{AD}_3$, $\text{AD}_4$, and $\text{AD}_5$). From these states, a second subunit can also desensitize, before the final step leading to the state in which all subunits are desensitized.

far the main contributors to the phenotypes in the dataset. This enabled us to reduce the model to ten different states, rather than thirty-four distinct states involving all subunits. We also simplified the activation transition whereby the resting receptor (R state) binds the agonist (AR state) and subsequently open (AO state). The model thus does not account for unliganded receptors openings (O state) that rarely occur at wild-type $\alpha 1 \beta 2 \gamma 2$ GABA$_A$Rs, with a spontaneous open probability as low as $10^{-5}$ in the absence of agonist[21], nor does it include the binding of two GABA molecules: we only considered the gating equilibrium for fully occupied receptors, as we work with supersaturating concentrations of GABA. From the AO state, either SU3, SU4, or SU5 can desensitize, to produce AD$_3$, AD$_4$, or AD$_5$ states, respectively. From these, the receptor can be further driven into states where two subunits are desensitized, e.g. desensitization of SU5 from the AD$_4$ state leads to the AD$_{45}$ state, where both SU4 and SU5 are desensitized. Finally, in that instance, SU3 could also desensitize to yield the AD$_{345}$ state, in which all three subunits are desensitized.

Using this general model, we progressively tuned the kinetic and functional parameters to best fit the dataset.

**Model I, in which desensitization of a single subunit shuts the channel, shows anti-synergistic behavior**. We first postulated that the receptor is functionally desensitized, i.e. non-conducting, as soon as one subunit is desensitized, with only the AO state allowing the passage of ions (Fig. 4a).

In model I, the desensitization and recovery rates ($\delta^+$ and $\delta^-$) for each subunit do not depend on the state of the other subunits. For simplicity, the parameters for SU4 and SU5 are set equal, since C$^4$ and C$^5$ display similar phenotypes. Thus, only four parameters ($\delta^+$, $\delta^-$, $\delta_3{}^+$, and $\delta_3{}^-$) are used to constrain the desensitization of C$^{WT}$, i.e. exactly the number of independent numerical constraints provided by the experimental data ($\tau_{fast}$, $\tau_{slow}$, %$A_{fast}$, %$I_{res}$). We also assumed that mutating subunit $i$ simply increases its desensitization rate by a ratio $c_i^+$ (Fig. 4b).

For each set of parameters, we performed kinetic simulations using QUB (Fig. 4 and Supplementary Table 3). Data are then analyzed using bi-exponential fitting of each virtual recording. In every simulation, we included all combinations of SU3, SU4 and SU5 mutants, from C$^{WT}$ to C$^{345}$.

In simulation "$a$", we set up the parameters to reproduce C$^{WT}$ and single mutant concatemers (Fig. 4c, d, f and Supplementary Table 3). However, these parameters largely underestimate the kinetics of the fast desensitization component for the double mutant C$^{45}$: simulation $a$ predicts a value of 1.76 s for the $\tau_{fast}$ of C$^{45}$, i.e. 10-fold slower than the experimental value. In simulation "$b$", we used the same parameters for C$^{WT}$, and set up the $c_i^+$ ratios to reproduce the C$^{45}$ phenotype (Fig. 4c, e, f and Supplementary Table 3). In that situation, we now largely overestimate the kinetics of the fast desensitization component for the single mutants C$^4$ and C$^5$: simulation $b$ predicts a value of 0.34 s for the $\tau_{fast}$ of both C$^4$ and C$^5$, i.e. an order of magnitude faster than the experimental values. In this particular example, it is striking that model I actually predicts anti-synergistic effects when mutating SU4 and SU5, with the fast desensitization kinetics of both the single and double mutants being similar (Fig. 4f).

Model I is thus incompatible with the dataset, and the reason is actually straightforward: if one desensitized subunit is enough to shut the pore, there should be a limiting fast subunit, whose mutation should have a strong effect on the kinetics of the fast desensitization component. This is not what we observe experimentally: the single mutant concatemers with the strongest phenotypes, C$^4$ and C$^5$, only display 40% increases in $\tau_{fast}$ (see above).

**Model II, in which at least two desensitized subunits are required to shut the pore, accounts for the synergy between SU4 and SU5 mutations**. We consequently modified the kinetic model to incorporate a key hypothesis: namely, that functional desensitization of the channel involves the rearrangement of at least two subunits, i.e. that AO, AD$_3$, AD$_4$, and AD$_5$ do conduct ions (model II, Fig. 5a).

To simulate responses with steady-state currents consistent with experimental values, we also allowed mutations to increase the rates for desensitization recovery of the mutated subunits (Fig. 5b; Supplementary Table 3). Indeed, not enabling this increase in recovery rates yields overestimated steady-state desensitization levels (Supplementary Fig. 4). Using this model II, we could perfectly account for the fast desensitization rate of C$^4$, C$^5$, and C$^{45}$ (Fig. 5c–e). When SU4 is mutated, SU5 desensitization still provides a limiting step for functional desensitization, acting as a brake, while in C$^{45}$ both "brakes" are relieved, enabling the channel to desensitize with fast kinetics, thereby generating a synergistic effect. This serves as a gentle reminder for studies using mutant-cycle analysis: it is indeed possible to have a strong functional coupling between non-interacting residues located far apart in a receptor's structure, if their motions are not concerted.

While model II accounts for the main features of the dataset, we further refined it to precisely fit some desensitization kinetics. Indeed, simulation of C$^3$ shows a mono-exponential process with %$A_{fast}$ = 100% (Fig. 5d, f), and an overestimated residual current (Fig. 5g; Supplementary Fig. 5). To circumvent this issue, we assumed that mutating SU3 increases the desensitization and recovery rates of SU4 (model II-$\beta$; Supplementary Fig. 6). From a structural point of view, such hypothesis seems plausible: the M3-5′ residue mutated in SU3 is located at the interface with SU4 (Fig. 1a, d–f), potentially interfering with conformational rearrangements of SU4. Using this model II-$\beta$, we could correctly simulate C$^3$ with two components for desensitization, (Supplementary Fig. 7a, c and Supplementary Table 3).

Still, for C$^{WT}$ and C$^3$, model II-$\beta$ produces an overestimation of both the fast component amplitude and the residual current (Supplementary Fig. 7c, d). Increasing the desensitization equilibrium constant ($\delta^+/\delta^-$) for SU4 and SU5 would reduce the amount of residual current, but would also lead to an increase in %$A_{fast}$ further out of the experimental range. Moreover, the rates of the slow desensitization components and the amplitudes of the fast components are both underestimated for C$^4$ and C$^5$, as well as for multiple mutant combinations (Supplementary Fig. 7b, c).

**Model III: adding inter-subunit coupling provides the best fit to experimental data**. We finally improved the model by adding a degree of structural coupling between adjacent subunits. We postulated that desensitization of a particular subunit would favor desensitization of its neighboring subunits. We thus incorporated coupling constants between subunit pairs in model III. The best fit was achieved assuming that, first, desensitization of SU4 decreases the recovery rate of SU5 by $\varepsilon = 10$-fold—and vice versa—and second that desensitization of SU4 increases the desensitization rate of SU3 by $\gamma = 100$-fold—and vice versa (Fig. 6a, Supplementary Table 3). Apart from these couplings, model III retains all features from model II-$\beta$ (Fig. 6a, b). Of note, we do not need to include any effect of SU4 or SU5 mutation on the recovery from desensitization (i.e. $c_4^- = c_5^- = 1$; Supplementary Table 3).

As shown in Fig. 7, model III largely accounts for experimental data, with experimental traces and simulated responses overlaying well (Fig. 7a–h), including for the wild-type situation. The fast

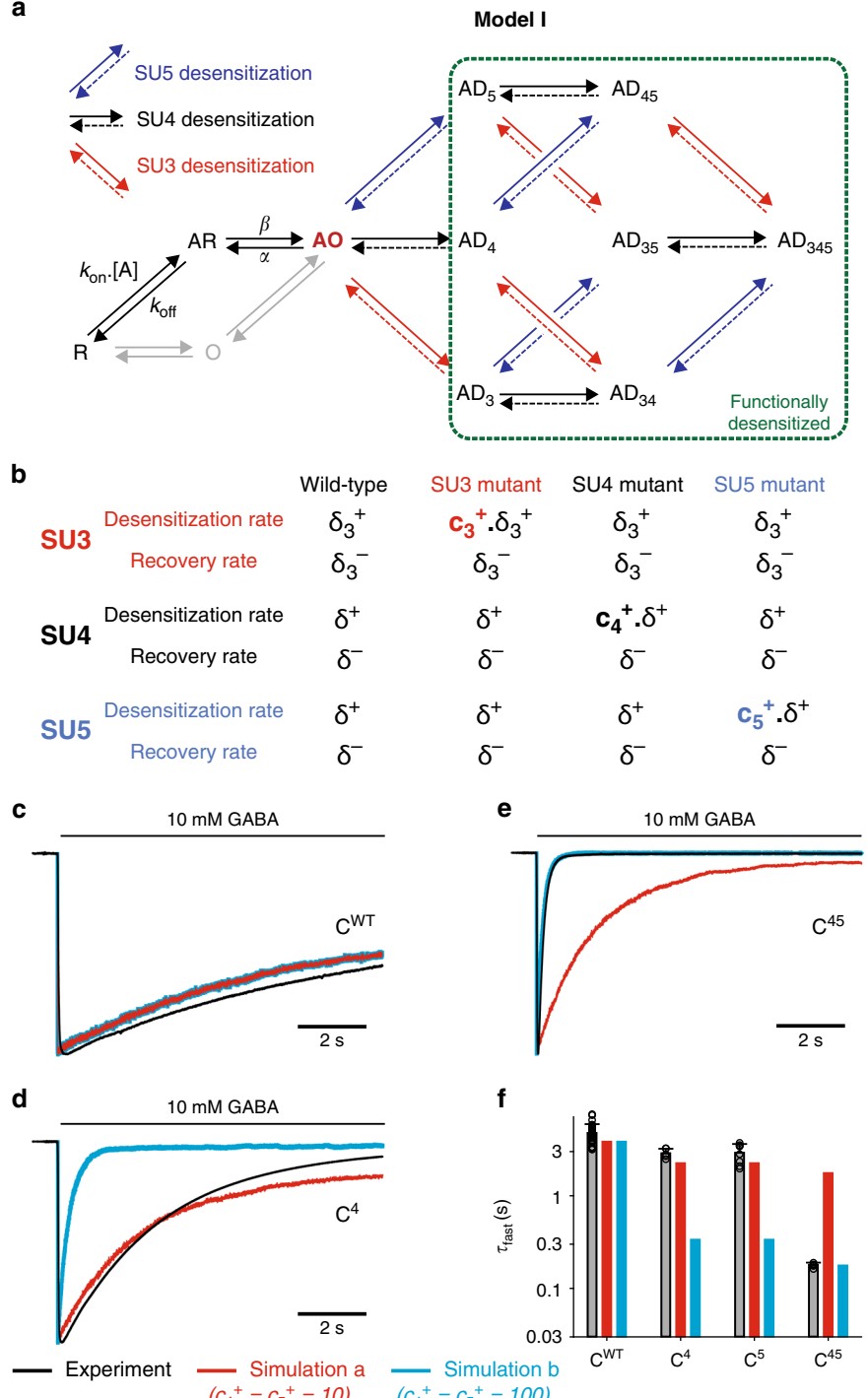

**Fig. 4 Model I: only the fully open state is conducting, and subunits move independently during desensitization. a** We assume in this model that a single desensitized subunit is enough to shut the pore of the channel, leading to functional desensitization. Moreover, subunits SU3, SU4, and SU5 can undergo a desensitization rearrangement independent of the other subunits. Thus, desensitization rates ($\delta_3^+$ for SU3, $\delta^+$ for SU4 and SU5) and recovery rates ($\delta_3^-$ for SU3, $\delta^-$ for SU4 and SU5) do not depend on the conformation of the neighboring subunits. **b** Effect of M3-5′ valine mutations in Model I. Mutations are hypothesized to specifically increase the desensitization rates of the mutated subunits, without altering any other parameter. **c–e** Representative currents for $C^{WT}$ (panel **c**), $C^4$ (panel **d**) and $C^{45}$ (panel **e**), in black, are compared to the outcome of two distinct simulations. In simulation *a* (red), the mutation-induced increase in the desensitization rates of SU4 and SU5 is adjusted so that the simulation of single mutants $C^4$ and $C^5$ broadly fits the experimental data, as seen in panel (**d**). In simulation *b* (blue), the mutation-induced increase in the desensitization rates of SU4 and SU5 is adjusted so that the simulation of the double mutant $C^{45}$ accounts for the experimental data, as seen in panel (**e**). **f** Bar graph summarizing the experimental data vs the predicted effects of SU4 and/or SU5 mutations on the kinetics of the fast desensitization component in simulations *a* and *b*. Experimental data are shown as means (bar graphs) and standard deviations (error bars), with individual data points indicated as circles. For panels **c–f** note that parameters from simulation *a* fail at describing the data for the double mutant $C^{45}$, while parameters from simulation *b* largely overestimate the effect of single mutants. See Supplementary Table 1 for numerical experimental values, the number of cells and number of independent series of experiments; and Supplementary Table 3 for the numerical values of parameters.

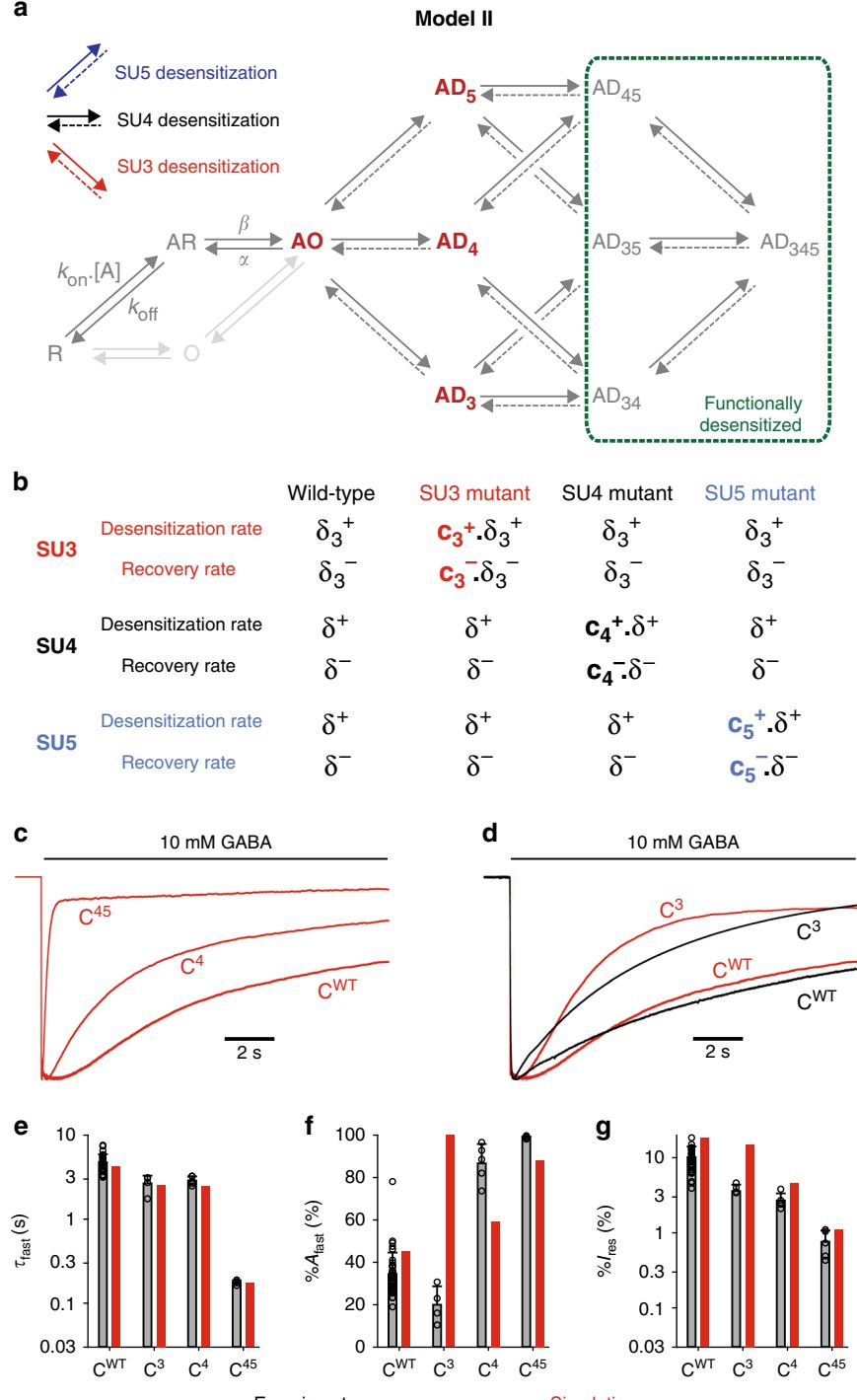

**Fig. 5 Model II: Two desensitized subunits are required to occlude the pore. a** Model II builds upon Model I by adding one key hypothesis: receptors with only one subunit in its desensitized conformation are still conducting, and desensitization occurs when at least two subunits are desensitized. Thus, states $AD_3$, $AD_4$, and $AD_5$ are open states from a functional point of view. **b** In Model II, mutation of a subunit can affect both its desensitization and recovery, as shown here with an example in which both SU4 and SU5 are mutated (construct $C^{45}$): $c_4^+$ and $c_5^+$ reflect the increase in desensitization rates, $c_4^-$ and $c_5^-$ reflecting the increase in recovery rates. **c** Simulated currents for $C^{WT}$, $C^4$ and $C^{45}$. **d** Representative currents for $C^{WT}$ and $C^3$ in black, are compared to their simulation counterparts in red. **e–g** Bar graphs summarizing the experimental data (in black) vs the simulations (in red) for the indicated concatemers on the kinetics (panel **e**) and the amplitude (panel **f**) of the fast desensitization component as well as the residual current after a 1 min long application of 10 mM GABA (panel **g**). Experimental data are shown as means (bar graphs) and standard deviations (error bars), with individual data points indicated as circles. Note that the results for the $C^5$ construct are not displayed, since the experimental data are almost identical to that of $C^4$ (see Fig. 2) and since the simulations for $C^4$ and $C^5$ are identical (see Supplementary Table 3). See Supplementary Fig. 5 for all simulation results; Supplementary Table 1 for numerical experimental values, the number of cells and number of independent series of experiments; and Supplementary Table 3 for the numerical values of parameters.

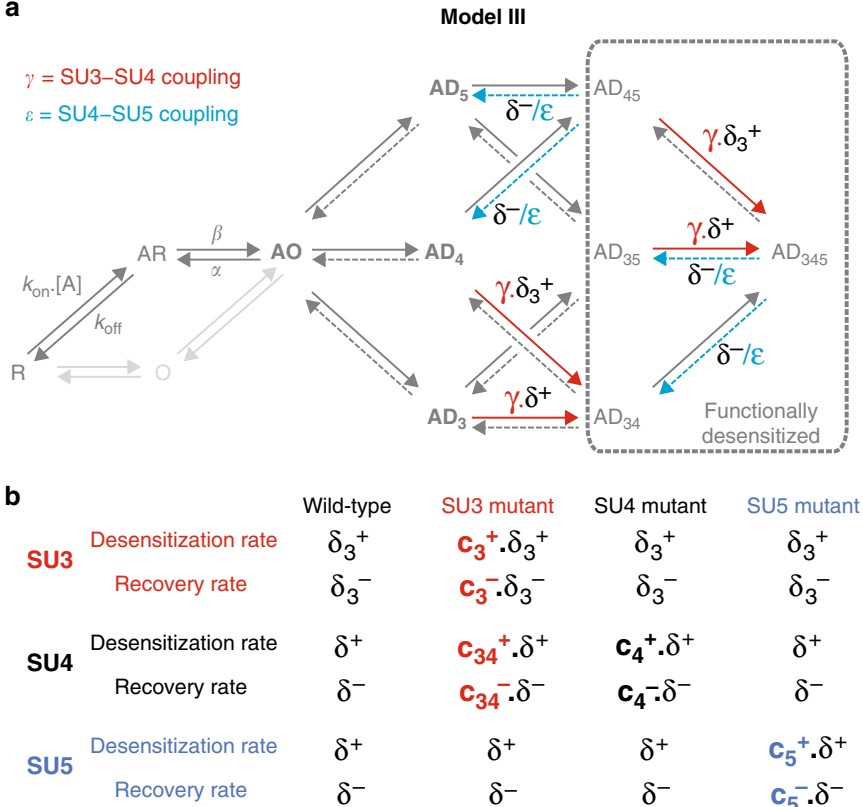

**Fig. 6 Model III introduces inter-subunit coupling during desensitization. a** For the wild-type receptors, Model III builds upon Model II by adding some coupling between adjacent subunits during desensitization. On the one hand, desensitization of SU3 accelerates the desensitization of SU4 by a factor $\gamma$, and reciprocally. On the other hand, desensitization of SU4 slows the recovery of SU5 by a factor $\varepsilon$, and reciprocally. **b** For mutated concatemers, Model III incorporates the additional hypothesis that the mutation of SU3 also affects the desensitization of SU4 by increasing both its desensitization and recovery rates, by ratios $c_{34}^+$ and $c_{34}^-$, respectively.

desensitization kinetics, which are the most reliable experimental constraints in the dataset, are particularly well simulated (Fig. 7i). The amplitudes of the fast component are overall in good agreement with the data, even though they are significantly underestimated for constructs $C^5$, $C^{34}$, and $C^{35}$ (Fig. 7k), while slow desensitization rates and steady-state currents are also underestimated for $C^{45}$ and $C^{345}$ (Fig. 7j, l). Those minor discrepancies might reflect the contribution of SU1 and/or SU2 to the receptors' desensitization, or even additional effects of the mutations (see "Discussion").

Altogether, the whole dataset is consistent with a non-concerted model for GABA$_A$Rs' desensitization, characterized by three main features: (1) subunits can rearrange one at a time during desensitization, the multiple temporal components of desensitization reflecting the existence of intermediate asymmetrical desensitized states; (2) rearrangements of adjacent subunits during desensitization are nonetheless partially coupled; and (3) the desensitization of at least two subunits is required to shut the pore, i.e. to lead to functional desensitization.

## Discussion

To illustrate the main features of our model of wild-type α1β2γ2 GABA$_A$Rs desensitization, we show in Fig. 8a the time-dependence of the various desensitized states' occupancies during desensitization. Since SU4 and SU5 desensitize the fastest, the receptors in the active state will transit first through a pre-desensitized open-pore state, in which either SU4 or SU5 is desensitized (Supplementary Fig. 8). Functional desensitization,

i.e. loss of electrophysiological response, subsequently occurs upon desensitization of the second fast subunit to yield the AD$_{45}$ state (Fig. 8a). The final step along the desensitization pathway would correspond to the desensitization of SU3, resulting in the slow component of desensitization, i.e. the entry in the AD$_{345}$ state (Fig. 8a). Like in all kinetic schemes where the slow- and fast-desensitized states are connected, this final step slowly depletes receptors from the fast-desensitized pool, which in turn displaces the overall population away from active conformations. We can thus extract the kinetically favored pathway and provide a schematic depiction of the movements of the M2 helices during desensitization, as shown in Fig. 8b. Interestingly, the requirement for two desensitized subunits to occlude the pore provides a framework to interpret results at α7 nAChRs, whose desensitization is blocked by PNU-120596. Indeed, at least four α7 subunits need to be bound by PNU-120596 in order to block desensitization, meaning that as soon as two subunits are unbound, the receptors can undergo functional desensitization[22]. We thus suggest that our kinetic scheme may be extended to the entire pLGIC family.

The whole dataset points to the γ2-subunit as a major determinant of the desensitization of α1β2γ2 GABA$_A$Rs. Interestingly, the γ2-TMD appears highly flexible in detergent conditions, collapsing within the pore when α1β3γ2 GABA$_A$Rs are solubilized in decylmaltoside neopentylglycol[23] or n-dodecyl-β-D-maltopyranoside[24,25]. The addition of lipids stabilizes the γ2-TMD in a more physiologically relevant conformation[23], but it still remains highly mobile and necessitates nanodiscs to be well resolved[17]. While the lack of the M3–M4 intracellular loop might

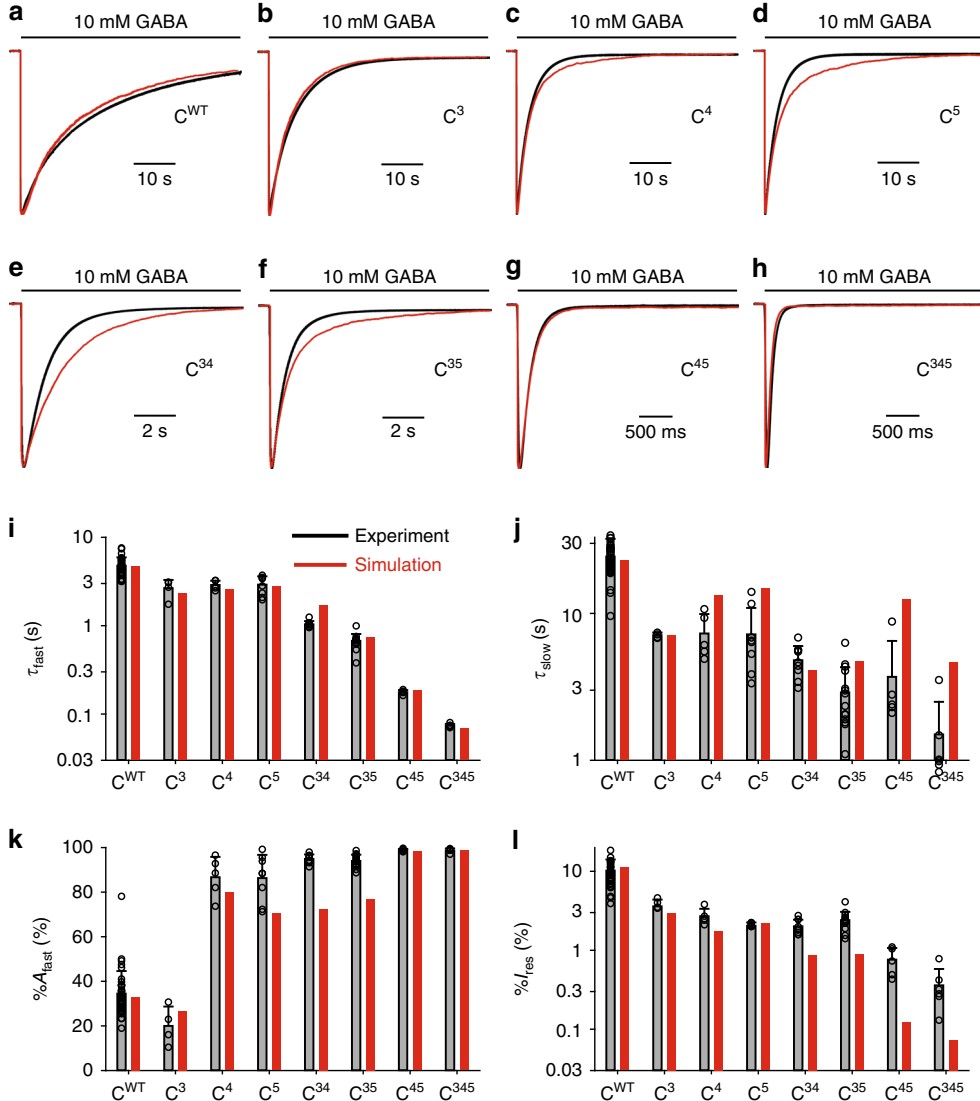

**Fig. 7 Model III simulations are broadly consistent with experimental data. a–h** Representative currents for the indicated constructs, in black, are overlaid with their simulation counterparts in red. Note the changes in timescales. **i–l** Bar graphs summarizing the experimental data (in black) vs the simulations (in red) for the indicated concatemers on the kinetics of the fast (panel **i**) and slow (panel **j**) desensitization components, the relative amplitude of the fast component (panel **k**) and the residual current after a 1 min long application of 10 mM GABA (panel **l**). Experimental data are shown as means (bar graphs) and standard deviations (error bars), with individual data points indicated as circles. See Supplementary Table 1 for numerical experimental values, the number of cells and number of independent series of experiments; and Supplementary Table 3 for the numerical values of parameters.

impact the structures solved in detergent, it is tempting to speculate that the dynamic nature of the γ2-TMD during desensitization is a functional counterpart of this biochemical instability. It is also interesting to note that the γ2-subunit contains a phosphorylation site at a serine located at the intracellular end of M3, namely S327[26]. This residue is located in an intracellular cassette modulating the desensitization properties of inhibitory pLGICs[8], eight residues downstream of the $M_3$−5′ residues that we have targeted in the current study. One could thus imagine that phosphorylation of γ2-S327 provides a mean to modulate the desensitization of γ2-containing GABA$_A$Rs. This would be consistent with a recent study showing that GABA$_A$Rs' desensitization promotes a form of long-term potentiation at inhibitory synapses by increasing the phosphorylation of γ2-S327[27]. Last but not least, the prominent role of the γ2-subunit in shaping the desensitization of α1β2γ2 GABA$_A$Rs makes it an interesting target for pharmacological modulation. Modulating

desensitization should barely affect basic synaptic signaling, potentially leading to fairly safe compounds with a large therapeutic window. Targeting the γ2-subunit specifically, in a desensitization locus with divergent sequences among pLGICs such as the intracellular end of the M3 segment, should also provide an efficient mean to achieve subtype selectivity. The current γ-selective pharmacology is embodied by the widely used class of benzodiazepines; unfortunately, benzodiazepines modulate GABA$_A$Rs likely by affecting a preactivation step, upstream from the ion channel opening[18]. They impact the overall conformational equilibrium of the ECD, as their binding affects indiscriminately both GABA binding sites[18], while desensitization per se most probably remains unchanged. Neurosteroids, which act at the transmembrane level and likely modulate desensitization[3], would be more promising, although their binding sites have currently been delineated for α and β subunits[28,29].

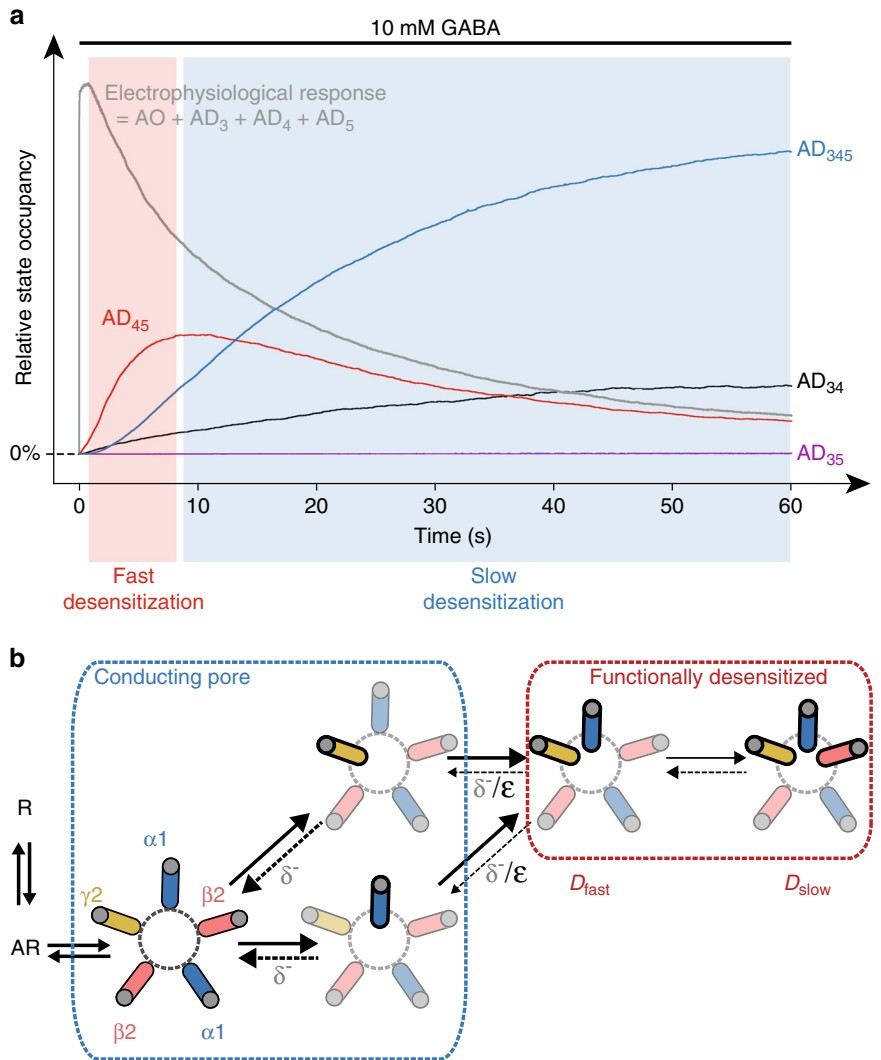

**Fig. 8 States occupancies predictions and structural depiction of Model III. a** The overall population of wild-type receptors in an active conformation is compared to the relative occupancies of the various desensitized states. As depicted by the red box, the early phase of desensitization is carried by the $AD_{45}$ state. On longer timescales (blue box), slow desensitization is largely embodied by the entry in the $AD_{345}$ state. The analysis of states occupancies was performed with QuB simulations. **b** In this simplified depiction of model III, we extracted the kinetically favored pathway for the desensitization of wild-type $\alpha1\beta2\gamma2$ $GABA_A$Rs. Upon agonist binding, the receptor is transiently stabilized in a fully open pseudo-symmetrical conformation. The two first subunits to rearrange during desensitization are the $\alpha1$ and the $\gamma2$ subunits involved in the binding of benzodiazepines, namely SU4 and SU5 in our concatemers. While one desensitized subunit is not enough to occlude the pore, fast desensitization corresponds to the rearrangement of both SU4 and SU5 subunits, which are coupled. Slow desensitization is then driven by the slower rearrangement of the SU3 subunit, i.e. the $\beta2$ subunit opposite to the $\gamma2$ subunit.

The apparent lack of effect on desensitization when mutating SU1 or SU2 alone is another striking feature of the dataset. A first hypothesis might be that these subunits do not desensitize during the one-minute-long GABA application. This is unlikely: in that case, mutating SU1 and/or SU2 should not affect the fast desensitization of concatemers harboring mutations on other subunits. However, mutating both subunits leads to an almost 2-fold increase in the fast desensitization kinetics of $C^{345}$ (Fig. 2; Supplementary Table 1). An alternative hypothesis would be that SU1 and/or SU2 display very-fast desensitization, but with a desensitization equilibrium largely displaced towards their open conformation ("$\delta^+/\delta^- \ll 1$"), thereby barely contributing to the macroscopic course of desensitization. Such desensitization equilibrium would minimally affect the size of currents, nor the apparent affinity for the agonist. In that event, it is conceivable that SU1 and/or SU2 mutations' effects could be revealed on a mutant background owing to inter-subunit coupling. This potential impact of inter-subunit coupling involving SU1 or SU2

might also explain why our kinetic simulations slightly differ from experiments for certain mutants—for example leading to an increased weight of the fast desensitization component ($\%A_{fast}$) of $C^{34}$ and $C^{35}$ as compared with our simulations. Such discrepancy could also be due to an effect of mutations on the inter-subunit couplings, with the SU3-SU4 coupling ($\gamma$) and the SU4–SU5 coupling ($\varepsilon$) being decreased by the SU3 and SU4 mutations, respectively. Our dataset unfortunately provides too little constraint to build a comprehensive scheme for these hypotheses, preventing their inclusion in our kinetic model. It is also worth stressing again that the experimental parameters driven by the slow component ($\%I_{res}$, $\tau_{slow}$) are difficult to measure reliably for strongly desensitizing mutants. In those cases, it is near impossible to fully discard the contribution of endogenous currents, or even the contribution of a tiny conductance from fully desensitized channels, as suggested for AMPA receptors[30]. One should thus be careful when interpreting such measurements—our most reliable measurements being the $\tau_{fast}$ values. It is noteworthy that

our experimental design allows for a 20–80% rise times in the 20–25 ms timescale. Therefore, very-fast desensitizing mutants may already desensitize during the onset of activation, compromising the accurate measurement of their fast desensitization component. Yet, we evaluate that our system allows for an accurate measurement of $\tau_{fast}$ down to the ~25 ms timescale (Supplementary Fig. 9), supporting that $\tau_{fast}$ has been correctly evaluated for all constructs used here.

Our non-concerted asymmetrical model provides a clear departure from a classical view in which $D_{fast}$ and $D_{slow}$ states are fundamentally different. It raises the possibility that these states are identical at the single-subunit level, with $D_{fast}$ only reflecting asymmetrical intermediates, mainly $AD_{45}$, along the desensitization process. Such scheme might appear surprising given the widely accepted concerted nature of pLGICs gating, as described for the muscle-type nAChR[31]. However, the analysis and concepts in favor of a concerted gating of pLGICs, like the MWC model framed more than half a century ago[32], have largely focused on biochemical and electrophysiological data obtained under gating equilibrium conditions such as concentration-response curves[21,31]. In the case of desensitization, the events are slow enough that intermediate events are directly detectable, namely the $D_{fast}$ state(s). If one could record the activation kinetics with sufficient temporal precision, it is likely that proper data fitting would also require the use of non-concerted asymmetric rearrangements. This is actually hinted by the prime model of muscle-type nAChR activation, in which conformational changes can affect independently either of the two ACh binding sites[33], as well as by rate-equilibrium free energy relationship analyses arguing for non-concerted rearrangements of M2-helices during nAChR activation[34]. Moreover, molecular dynamics studies also pinpoint the cytoplasmic end of the pore as a locus for asymmetric conformations at the μs-timescale: the five −2′ M2-residues are often distributed in a non-symmetrical fashion during simulations of the open state of the zebrafish α1 Glycine receptor[35,36]. Of note, channels and receptors from other families are also known to rely on asymmetric gating. This is the case of the prokaryotic magnesium channel CorA, whose active state actually stems from an asymmetric conformation as reported by cryo-electron microscopy[37]. This is also the case for NMDA receptors, for which the cryo-electron microscopy of tri-heteromeric GluN1/GluN2A/GluN2B receptors reveals an asymmetric organization[38].

The exact structural underpinnings of desensitization remain however ill-defined, in particular since the current structures have been obtained for presumable resting and desensitized conformations so far[10,17]. In the absence of an active conformation, one can only speculate on the precise molecular events occurring during the active to desensitized transition.

## Methods

**Molecular biology.** The GABA$_A$ concatemeric α1β2γ2 construct was previously described[18], based on the concatenation of mouse GABA$_A$ subunits. Briefly, the five subunits were subcloned in the order β2–α1–β2–α1–γ2 into a low copy number vector pRK5, retaining the peptide signal of the first subunit only. We used the short splice variant of the γ2 subunit, γ2S. All five subunits are flanked by unique restriction sites to allow the subcloning of mutated subunits, and separated by 15–20 residues long polyglutamine linkers, depending on the length of the C-terminus end of the subunit preceding the linker. The construct thus shows the arrangement ClaI-β2-20Q-AgeI-α1-15G-SalI-β2-20Q-NheI-α1-15Q-γ2S-Stop-HindIII. Site-directed mutagenesis was performed on individual subunits as previously described[8]. Owing to the unique restriction sites, mutated subunits were then sequentially subcloned in the concatemer to yield the desired combinations of mutated subunits. We finally sequenced the resulting mutated concatemers to check for the incorporation of the desired mutated subunits. We could not use primers annealing anywhere in α1 or β2 for sequencing, as both subunits are present as duplicates in the concatemer. Instead, we sequenced SU1-4 subunits with primers annealing at their 5′ DNA extremity, centered on the sequence of the unique restriction site preceding the following subunit. Such reverse primers enable the sequencing of the 5′ end of the subunits' DNA, coding for their C-terminus once translated.

**Expressing GABA$_A$Rs in Xenopus laevis oocytes.** Ovaries from *Xenopus laevis* were obtained from CRB Xenopes in Rennes. Free oocytes were obtained by incubating segments of ovary in collagenase type 1 (Sigma) dissolved in a Ca$^{2+}$-free OR2 solution, which contained (mM): 85 NaCl, 5 HEPES, 1 MgCl$_2$, pH adjusted to 7.6 with KOH. After 2-4 h exposure to collagenase I, defolliculated oocytes were washed several times with OR2, and thereafter maintained in a Barth's solution containing (mM): 88 NaCl, 1 KCl, 0.33 Ca(NO$_3$)$_2$, 0.41 CaCl$_2$, 0.82 MgSO$_4$, 2.4 NaHCO$_3$, 10 HEPES, pH adjusted to 7.6 with NaOH. Single oocytes were injected with 27.6 nl of concatemeric GABA$_A$R cDNAs (nuclear injection) at a concentration of 30 ng/μl. Oocytes were incubated at 17 °C in Barth's solution devoid of serum or antibiotics.

**Two-electrode voltage clamp recording.** Oocytes expressing pentameric concatemers were recorded 2-4 days after injection. They were superfused with a solution containing (mM): 100 NaCl, 2 KCl, 2 CaCl$_2$, 1 MgCl$_2$, 5 HEPES, pH adjusted to 7.4 with NaOH. Solution flowed at an approximate speed of 12 mL/min. Currents were recorded using a Warner OC-725C amplifier, a Digidata 1550 A interface and pCLAMP 10 (Molecular Devices). Currents were digitized at 500 Hz and filtered at 100 Hz (30–60 Hz used for display purposes). Oocytes were voltage-clamped at −60 mV and experiments conducted at room temperature. Desensitizing currents were induced by 1 min applications of 10 mM GABA. 20–80% current rise times of 20–25 ms were achieved for $C^{WT}$.

**Data analysis.** The extent of desensitization was determined as $(1−I_{res}/I_{peak})$, where $I_{peak}$ is the peak current and $I_{res}$ the residual current remaining at the end of the agonist application. Weighted decay time constants for desensitization were determined by fitting the desensitizing phase with two exponential components (pCLAMP 10.6.0.13), as given by the following equation: $\tau_w = \%A_{fast} * \tau_{fast} + (1−\%A_{fast}) * \tau_{slow}$. All data values are means ± standard deviation.

**Drugs and chemicals.** All compounds were purchased from Sigma. GABA was prepared as a 1 M stock solution in recording solution. Aliquots were stored at −20 °C.

**Kinetic modeling.** We used QUB[20] (QUB Express 1.12.6 and QUB online) to build Markov-chain kinetic models. Each simulation contained 10,000–30,000 channels. The binding and gating rate constants are broadly consistent with previously published values for GABA$_A$Rs[39]. Except for Supplementary Fig. 9, the simulation protocol consisted in a step application of 10 mM GABA (instantaneous concentration change). For each model, we performed iterative rounds of kinetic simulations by adjusting manually the set of parameters. Binding and gating constants being fixed, Model I (Fig. 4), Model II (Fig. 5), Model II-β (Supplementary Fig. 6), and the concerted model (Supplementary Fig. 3) only contain four parameters for the wild-type receptors ($\delta^+$, $\delta^-$, $\delta_3^+$, and $\delta_3^-$ for Models I, II and II-β; fast and slow desensitization rates and their recovery counterparts for the concerted model). This equates to the number of independent experimental measurements related to the two desensitization components ($\tau_{fast}$, $\tau_{slow}$, $\%A_{fast}$, $\%I_{res}$). We could thus be confident that, once we have a set of parameters accounting for the wild-type data, the model has a good predictive value. We used ballpark figures to build the initial set of parameters, already having in mind that the fast desensitization component might be mostly carried by subunits 4 and 5 (due to the $C^{45}$ phenotype). For example, taking into account only the two pathways linking the AO state to the $AD_{45}$ state in Model I, the fast desensitization kinetics could be approximated with $\tau_{fast} \sim 2.(\delta^++\delta^-)$, while the amplitude of the fast component would yield estimates for the ratio $D = \delta^+/\delta^-$ approximated with the equation $A_{fast}/I_{peak} \sim 2.D$, resulting in a D value approximated by ~ 0.2, as well as $\delta^+$ and $\delta^-$ values of ~ 0.2 s$^{-1}$ and 1 s$^{-1}$, respectively. In Model II, $\tau_{fast}$ is in the order $2.(\delta^+/\delta^-).\delta^+$, while the ratio D is constrained by the fast component amplitude ($A_{fast}$) with the following approximation: $A_{fast}/I_{peak} \sim D^2/(1 + 2.D)$. Such approximation yields D ~ 0.9 and $\delta^+$ ~ 0.14 s$^{-1}$. Mutation-induced changes in those parameters ($c_3^+$, $c_3^-$, $c_4^+$, $c_4^-$, $c_5^+$, $c_5^-$) for models Models I, II were then adjusted manually to account for the effects of individual mutants ($C^3$, $C^4$, and $C^5$), Model II-β requiring the additional adjustment of $c_{34}^+$, $c_{34}^-$ for the effect of SU3 mutation. The effects of mutations in the concerted model (Supplementary Fig. 3) led to four mutation-related parameters ($\gamma_f$, $\varepsilon_f$, $\gamma_s$, and $\varepsilon_s$) for each individual mutant $C^3$, $C^4$, and $C^5$. In all cases, mutation-related parameters derived from individual mutants were then combined to predict the effect of combining and mutations in constructs $C^{34}$, $C^{35}$, $C^{45}$, and $C^{345}$. The quality of the fit was merely assessed by visual inspection of the bar graphs illustrating the predictions for $\tau_{fast}$, $\tau_{slow}$, $\%A_{fast}$, and $I_{res}$. It may thus be possible to obtain better fits to the data. For Model III, we generated a series of wild-type models with values for coupling constants ($\gamma$ and $\varepsilon$) in the 1–1000 range (1, 10, 100, and 1000). We next manually adjusted the mutation-induced changes as described above for Model II-β.

**Statistics and reproducibility.** As an internal quality control, for each batch of *Xenopus laevis* oocytes used to express mutant concatemers, we recorded some oocytes expressing the wild-ype concatemers, thereby ensuring we could replicate recordings consistent with overall data for wild-type concatemers. For each construct, we performed at least 2 series of independent experiments (oocytes obtained

from ovaries of two different animals), and recorded at least 2 cells for each series of recordings, yielding a total of at least 4 cells. See Supplementary Table 1 for the exact number of cells and animals used for each construct. All attempts were successful, i.e. each series of *Xenopus* oocytes DNA injection yielded experimental data used in the present work.

**Reporting summary**. Further information on research design is available in the Nature Research Reporting Summary linked to this article.

## Data availability

Data supporting the findings of this manuscript are available from the corresponding author upon reasonable request. A reporting summary for this Article is available as a Supplementary Information file. Source data are provided with this paper.

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

## Acknowledgements

The authors would like to thank Drs Thomas Boulin, Hugues Nury, Laurie Peverini, Marie Prevost and Prof Trevor Smart for critical reading of the manuscript; and acknowledge financial support by the Fondation de la Recherche Médicale (grant "Équipe FRM" DEQ20140329497 to P.-J.C.) and the European Commission Research Executive Agency (Marie Sklodowska-Curie Action, Individual Fellowship 659371 to M.G.; ERC Advanced Grant GA788974 Dynactinote to P.-J.C.). M.G. is grateful to the Fondation Bettencourt Schueller for their support.

## Author contributions

M.G. designed the study; M.G. and N.B. performed molecular biology and TEVC recordings; M.G. analyzed the data and performed the kinetic modeling; all authors discussed results; M.G. and P.-J.C. wrote the manuscript; P.-J.C. and M.G. acquired funding.

## Competing interests

The authors declare no competing interests.
