## [Peer Review File · Nature Communications]

REVIEWER COMMENTS

Reviewer #1 (Remarks to the Author):

This study by Gielen et al. examines the structural mechanism of GABA receptor desensitization through mutagenesis, electrophysiology and kinetic modeling. This work builds from (recently) a couple key papers from Gielen, Corringer and Smart on this larger topic of desensitization across the pentameric channel family. Here the authors leverage a mutation in the M3 helix that provides a gain of function in accelerating desensitization and increasing its amplitude. They mutate this M3 residue in each of the $\alpha 1$, $\beta 2$ and $\gamma 2$ subunits and perform TEVC experiments with relatively fast solution exchange to determine tau-fast, tau-slow and the %amplitude of current decay from each component. They test all combinations of mutant subunits in the tri-heteromeric receptor. They then generate kinetic models which they use to interrogate concerted vs. non-concerted desensitization mechanisms. They find strikingly different contributions from subsets of subunits to desensitization, and differential coupling among subunits in the pentameric ring. Overall this study is elegant, thoughtful, and clearly written. Its significance is two-fold. First, the structural underpinnings of pentameric channel desensitization are not well understood, but they fundamentally shape synaptic and extrasynaptic excitability. Second, this study provides at least one robust example of the concerted allosteric model not holding true, which is a major finding. It reveals desensitization occurring at the subunit level in an asymmetric manner, with two subunits needing to adopt the single desensitized conformation in order to occlude the permeation pathway. This finding suggests that the fast and slow desensitized states differ only in the number of subunits adopting a desensitized conformation. Fascinating work that will provide fodder for additional studies. I find this work appropriate for Nature Communications with minor suggestions for revision.

1. Prime numbering scheme for M3 is not intuitive as presented; pentameric channel people will be familiar with that scheme for M2, but if you want to use it for M3, some explanation of where the numbering starts would be helpful.
2. It makes good sense as discussed on p. 8 that for the more rapidly desensitizing mutants some of the peak response may not be observed. Given the stated 20-25 ms rise times for the 20-80% response, how fast does the tau-fast for desensitization need to be before it cannot be accurately measured with this system?
3. In the paragraph preceding the Discussion, you suggest that this non-concerted model is relevant across the pLGIC superfamily, citing no data. Maybe wait to make this conclusion until you have discussed the alpha7 + PNU findings?
4. P. 16-17, on the topic of $\gamma 2$ subunit and its role in desensitization- is there pharmacology that could be tied into this idea, specific ligands that bind at a $\gamma 2$ interface that affect desensitization kinetics? I do not suggest doing new experiments, just incorporating existing literature to broaden the scope. Benzodiazepines, barbiturates, perhaps volatile anesthetics?
5. Why were the recovery rates allowed to be affected by mutations in model 2, but not model 1?
6. In figure panel 7K, simulations of mutants C34 and C35 deviate from the experimental data considerably. Particularly in the case of C34, where adjacent (and therefore coupled) subunits are mutated, one would expect the %Afast to be very high. Do you have some insights into why the model and the experimental results differ here?
7. Is there any chance the location of the linkers may affect the effects of mutations?

Reviewer #2 (Remarks to the Author):

The manuscript by Gielen et al reports a quantitative and in-depth study of desensitization mechanism in GABAA receptor. The authors use a tandem construct of $\alpha 1\beta 2\gamma 2$ heteromeric GABAA receptor to engineer gain-of-function mutation at a conserved position, in one subunit at a time, and in multiple combinations of subunit with the goal of dissecting the contribution of each subunit to the overall

desensitization process of the channel. Macroscopic currents were recorded by TEVC in oocytes, desensitization rates from bi-exponential fits to currents decays were used to develop kinetic models, and in turn these models were used to simulate current traces and compared to the experimental traces. The authors conclude that pore closure during desensitization involves asymmetric and independent contributions from the different subunits, and that the pore occlusion may require movement of at least two subunits, and with some level of cooperativity between adjacent subunits.

Overall, clever design of experiments and a well-executed study. The authors do a nice job with the discussion of the model, the physical interpretation, and the physiological significance. However, there are several areas where details are lacking or the authors could provide more clarification.

- The study involves a quantitative modeling of current decays from TEVC upon ligand-perfusion. Given the need for sensitivity in measurements of the individual components of the decay, it is not clear why the authors have chosen this system rather than whole-cell measurements in HEK cells with fast ligand application.
- Related to the above question, the time constant for fast component of C12345 is in the ~ 30 ms range. Is it not a concern that these rates are at the limit of solution exchange rates? The peak current amplitude of this mutant is among the lowest (~ 0.3 microA), and may even suggest the channels are desensitizing prior to opening.
- "...do not alter significantly the concentration-response curve of the GABA-elicited peak currents measured before the onset of desensitization. This indicates only a weak effect of the mutations on the resting-to-active state transition, and a major effect on the active-to-desensitized state transition.." Not clear how the authors say this conclusively from EC50 derived from macroscopic currents. Also, wouldn't it be expected to see a left shift in dose response for gain-of-desensitizing mutants.
- It is surprising that the authors use QUB to simulate currents from the various models but not use these models to directly fit the measured currents and estimate the parameters.
- What is the rationale for the initial parameters (d^- , d^+ , $d3^+$, $d3^-$) used in the model. Details of how the fine tuning of the kinetic parameters done? How was the quality of the fit assessed? The models include recovery rates from D_{fast} and D_{slow} . Recovery is not independently determined in this study, but estimated from the decay. Are there evidence in literature that the mutation does not affect recovery? The issue is for fast desensitizing constructs, the slow decay component is underestimated and hence the on and recovery rates are uncertain.
- Occurrence of multiple "Pre-sensitized open state" at same conductance level is assumed. Is there any evidence from single-channel studies, perhaps with difference in mean open times?

Sudha Chakrapani

REVIEWER COMMENTS

We thank the reviewers for their constructive comments, and are glad to learn that they found our study of interest.

Prior to answering the reviewers' specific comments, we want to indicate the following updates and minor corrections:

- We realized there was a mistake in the bar graph values for C^{35} in Model 2 (former Suppl. Figure 3). We have corrected them in Suppl. Figure 5.
- We have added 6 new references (4 in the main text, 2 in the supplementary text. One reference is a new study by the Smart Lab, discussing the effect of GABAAR desensitization on a new form of LTP, mediated by the phosphorylation of $\gamma 2$ -S327 (currently in bioRxiv). Another reference, from the Cull-Candy lab (published last year in Nat Commun), suggests that desensitized AMPA receptors actually display a very small conductance (not equal to zero), is included to discuss current assumptions (and potential limitations) in our model. Two other references have been added to discuss the binding sites of neurosteroids at GABAARs. In supplementary discussion, a reference (in our Suppl. Discussion) from the Wollmuth lab (currently in bioRxiv), highlights the importance of using non-equilibrium conditions (in their case, fast applications on outside-out patches) to examine fine gating details, which would not be accessible using cell-attached recordings. The last reference (Olsson, U., J. Stat. Educ., 2005) is only to explain how we calculated 95% confidence intervals for the distribution of peak current amplitudes (as shown in the updated Suppl. Table 1). Finally, we have updated the reference to the latest paper from the Chakrapani lab on the desensitization of GlyRs (now published in Nat Commun).
- As requested in the reporting summary, we now display the individual data points used to build Figure 2. These are shown in the new Supplementary Figure 1.

Reviewer #1 (Remarks to the Author):

This study by Gielen et al. examines the structural mechanism of GABA receptor desensitization through mutagenesis, electrophysiology and kinetic modeling. This work builds from (recently) a couple key papers from Gielen, Corringer and Smart on this larger topic of desensitization across the pentameric channel family. Here the authors leverage a mutation in the M3 helix that provides a gain of function in accelerating desensitization and increasing its amplitude. They mutate this M3 residue in each of the $\alpha 1$, $\beta 2$ and $\gamma 2$ subunits and perform TEVC experiments with relatively fast solution exchange to determine tau-fast, tau-slow and the %amplitude of current decay from each component. They test all combinations of mutant subunits in the tri-heteromeric receptor. They then generate kinetic models which they use to interrogate concerted vs. non-concerted desensitization mechanisms. They find strikingly different contributions from subsets of subunits to desensitization, and differential coupling among subunits in the pentameric ring. Overall this study is elegant, thoughtful, and clearly written. Its significance is two-fold. First, the structural underpinnings of pentameric channel desensitization are not well understood, but they fundamentally shape synaptic and extrasynaptic excitability. Second, this study provides at least one robust example of the concerted allosteric model not holding true, which is a major finding. It reveals desensitization occurring at the subunit level in an asymmetric manner, with two subunits needing to adopt the single desensitized conformation in order to occlude the permeation pathway. This finding suggests that the fast and slow desensitized states differ only in the number of subunits adopting a desensitized conformation. Fascinating work that will provide fodder for additional studies. I find this work appropriate for Nature

Communications with minor suggestions for revision.

1. Prime numbering scheme for M3 is not intuitive as presented; pentameric channel people will be familiar with that scheme for M2, but if you want to use it for M3, some explanation of where the numbering starts would be helpful.

We now add a brief description of the M3 prime numbering on page 6 (“this prime notation, akin the one largely used for the M2 segment, starts at the cytoplasmic end of the M3 segment”), as described in ref 19 (Jaiteh, M., Taly, A. & Héning, J. Evolution of Pentameric Ligand-Gated Ion Channels: Pro-Loop Receptors. PLOS ONE 11, e0151934 (2016)).

2. It makes good sense as discussed on p. 8 that for the more rapidly desensitizing mutants some of the peak response may not be observed. Given the stated 20-25 ms rise times for the 20-80% response, how fast does the tau-fast for desensitization need to be before it cannot be accurately measured with this system?

To address this issue, we have performed a series of simulations to examine the influence of the 20-25 ms rise times (20-80% response) as compared to the instantaneous solution exchanged (used in all simulations) on the apparent fast desensitization kinetics. To this aim, we used a simplified kinetic scheme for WT receptors (Supplementary Figure 9 panel A, based on figure 8). We modeled the agonist perfusion either as an instantaneous step stimulation, or as a linear increase in concentration of GABA from 0 to 10 mM in 250ms, a solution exchange called ramp stimulation that yields a rise time of ~ 23 ms (Supplementary Figure 9 panel B).

Panel C shows that the apparent desensitization kinetics of the WT receptor is nearly identical following step or ramp stimulation, consistent with its slow onset of desensitization. We then modeled what would happen to mutant receptors with increased desensitization kinetics (mutants A-F), whose fast desensitization decay time range from 511 ms down to 7.1 ms with a step stimulation. As can be seen on panels D-I (raw simulations) and panel J (bar graph displaying τ_{fast} analysis), the peak response starts being affected for decay times briefer than ~ 40 ms with the ramp stimulation (Mutant C, panel F). However, the fast desensitization kinetics remain accurate in the ramp stimulation procedure up to the point where $\tau_{\text{fast}} \sim 20$ ms. Indeed, for Mutant D, $\tau_{\text{fast}} = 25.7$ ms measured in step stimulation is close to $\tau_{\text{fast}} = 27.2$ ms measured in ramp stimulation. On the contrary, for Mutant E, $\tau_{\text{fast}} = 12.9$ ms measured in step stimulation deviates from $\tau_{\text{fast}} = 18.7$ ms measured in ramp stimulation, a deviation that is further increased for mutants displaying faster kinetics.

We now incorporate this analysis in the manuscript (see Discussion p. 19 and Suppl. Figure 9).

3. In the paragraph preceding the Discussion, you suggest that this non-concerted model is relevant across the pLGIC superfamily, citing no data. Maybe wait to make this conclusion until you have discussed the alpha7 + PNU findings?

Thanks for the suggestion, this has been done.

4. P. 16-17, on the topic of g2 subunit and its role in desensitization- is there pharmacology that could be tied into this idea, specific ligands that bind at a g2 interface that affect desensitization kinetics? I do not suggest doing new experiments, just incorporating existing literature to broaden the scope. Benzodiazepines, barbiturates, perhaps volatile anesthetics?

This is indeed an interesting point regarding pharmacology, although we speculate that ligands binding at the ECD (e.g., benzodiazepines) will likely affect the overall conformation of the ECD (and thus the preactivation steps) while leaving unaffected the desensitization process *per se*. Ligands acting at the TMD such as Neurosteroids are probably the most interesting target for this kind of modulation, although the current view is that their binding sites mostly target α and β subunits. We hope our study will foster the rational design of γ -selective neurosteroid-derived modulators. We now briefly discuss this in the main text (discussion p17-18).

5. Why were the recovery rates allowed to be affected by mutations in model 2, but not model 1?

We omitted an intermediate model 2', in which recovery rates are unaffected, in order not to flood the reader with too many models, but obviously this requires justification (even though the changes in recovery rates are moderate – the final model actually does not incorporate any effect of mutating subunit 4 or 5 on the recovery rates). We now include the model 2' in Supplementary Figure 4 (referenced on page 13 of the main text). With model 2', the extent of desensitization was largely overestimated (very low I_{res}), and this is the reason why we implemented the increases in recovery rates.

6. In figure panel 7K, simulations of mutants C34 and C35 deviate from the experimental data considerably. Particularly in the case of C34, where adjacent (and therefore coupled) subunits are mutated, one would expect the %A_{fast} to be very high. Do you have some insights into why the model and the experimental results differ here?

The reviewer is right in pointing out that %A_{fast} from our model, in particular, deviate from the experimental data for C34 and C35. Several hypotheses might explain such a discrepancy:

- First, our model does not include SU1 and SU2, and it remains possible that these subunits play a role on the macroscopic course of desensitization when other subunits are mutated (especially if taking into account a potential coupling between SU2 and SU3, as well as between SU5 and SU1). We mentioned this possibility in the text and now emphasize it more (see Discussion p. 18-19).

Second, the intersubunit couplings themselves might be affected by mutations. For example, we could expect the SU3-SU4 coupling (γ) to be decreased by the SU3 mutation, while the SU4 mutation might decrease the coupling between SU4 and SU5. However, we don't feel comfortable about adding such modifications in our model, since this will add new parameters without enough constraints from experimental data to draw firm conclusions. We now discuss this point in the manuscript (see Discussion, p. 18-19).

Last, our model provides a template for analyzing the receptors' desensitization assuming that desensitized channels are fully shut (single channel current = 0 pA). We cannot fully discard the possibility that desensitized channels carry a very low amount of current, as has been suggested for AMPA receptors (see ref. Coombs, ..., Cull-Candy, Nat Commun, 2019), or even that a tiny current in our recordings is carried by endogenous channels. In such case, our experiments might over-estimate the number of fully active channels under steady state conditions, which would bias our modeling

– including the amount of intersubunit coupling, affecting in turn the amplitude of the fast desensitization component (see Discussion, p. 18-19).

7. Is there any chance the location of the linkers may affect the effects of mutations?

The reviewer raises an important point here, which deserves more explanation.

The linkers are located in the extracellular part of the receptor, connecting the extracellular C-terminal end of the preceding subunit to the N-terminal end of the following subunit's ECD. This location of course prevents any direct interaction with the mutated residues, located at the intracellular end of the pore.

If the linkers were to constrain the receptors' conformations, one would predict that the first order effect would be to constrain the conformational dynamics of the ECD, which would in turn impact on the pharmacology of extracellular ligands, including GABA (binding at the β - α extracellular interface) and benzodiazepines (binding at the α - γ extracellular interface). The fact that the apparent affinity for GABA and the modulation by benzodiazepines are similar for receptors assembled from loose subunits and for our pentameric concatemer (See Gielen et al., J Neurosci, 2012) thus argues again a significant effect of the linkers on the receptors' gating.

In addition, comparing our published data (Gielen et al., Nat Commun, 2015) with that of the present study, the desensitization kinetics and extent from $\alpha 1\beta 2\gamma 2$ receptors expressed from loose subunits are similar, if not identical, to the ones from our pentameric construct – both for the wild-type receptors, as well as for the $\gamma 2$ -H318V single mutant. This again provides evidence that the linkers do not apply a significant constrain on the conformational changes at play during desensitization.

We now discuss these points in the manuscript (see pages 5, 6 and 7)

Reviewer #2 (Remarks to the Author):

The manuscript by Gielen et al reports a quantitative and in-depth study of desensitization mechanism in GABAA receptor. The authors use a tandem construct of $\alpha 1\beta 2\gamma 2$ heteromeric GABAA receptor to engineer gain-of-function mutation at a conserved position, in one subunit at a time, and in multiple combinations of subunit with the goal of dissecting the contribution of each subunit to the overall desensitization process of the channel.

Macroscopic currents were recorded by TEVC in oocytes, desensitization rates from bi-exponential fits to currents decays were used to develop kinetic models, and in turn these models were used to simulate current traces and compared to the experimental traces. The authors conclude that pore closure during desensitization involves asymmetric and independent contributions from the different subunits, and that the pore occlusion may require movement of at least two subunits, and with some level of cooperativity between adjacent subunits.

Overall, clever design of experiments and a well-executed study. The authors do a nice job with the discussion of the model, the physical interpretation, and the physiological significance. However, there are several areas where details are lacking or the authors could provide more clarification.

- The study involves a quantitative modeling of current decays from TEVC upon ligand-perfusion. Given the need for sensitivity in measurements of the individual components of the

decay, it is not clear why the authors have chosen this system rather than whole-cell measurements in HEK cells with fast ligand application.

This is indeed an important point, on which we have elaborated on in a recent review paper (Gielen & Corringer, J Physiol, 2018, see « Box 1. The variability of desensitization »).

Our case for using TEVC recordings of *Xenopus* oocytes is based on the following arguments:

- In outside-out patches (be it from HEK cells or *Xenopus* oocytes), desensitization kinetics (and extent) of most pLGICs, including that of GABA_AR, are extremely variable from one patch to another.
- In whole-cell recordings of HEK 293 cells, time-dependent increase in desensitization kinetics (and extent) is consistently observed in the course of the recording, suggesting a time-dependent dialysis effect of the intracellular compartment, or redistribution of lipids like PIP2 due to patch clamping.
- Even when considering only the « slow » components of desensitization, the data from HEK whole-cell, HEK outside-out or *Xenopus* outside-out recordings are not consistent with TEVC recordings, which in contrast are far more reproducible (thus enabling the current study)
- Last but not least, if one considers that TEVC currents only reflected the « steady-state » component of patch-clamp experiments, there should be an increase in the apparent affinity for GABA in TEVC recordings compared to patch-clamp experiments. The reverse is actually true, since the EC₅₀ for GABA activation in TEVC recordings is about 10-fold higher than the one measured in patch-clamp.

To conclude, we cannot be fully affirmative that TEVC recordings of *Xenopus* oocytes accurately recapitulates the neuronal condition (including lipid bilayer composition) for GABAARs' desensitization. However, it is unclear whether patch-clamp recordings (from HEK cells or *Xenopus* oocytes) would be more relevant in this regard.

More importantly, results with TEVC recordings are reproducible enough to enable the present study, unlike patch-clamp recordings, which we now remind on page 6 (“As discussed in a previous publication, TEVC recordings of *Xenopus laevis* oocytes are well-suited to the study of desensitization of pLGICs owing to the robustness of the approach, which contrasts with the very high inter- and intra-cellular variability when recording desensitization with patch-clamp methods.”).

Our aim is here not to provide the reader with the most accurate and physiologically-relevant desensitization rates (that are likely dependent on the expression system), but rather to highlight desensitization mechanisms that are intrinsic to the GABA_A receptor molecule.

- Related to the above question, the time constant for fast component of C12345 is in the ~ 30 ms range. Is it not a concern that these rates are at the limit of solution exchange rates? The peak current amplitude of this mutant is among the lowest (~0.3 microA), and may even suggest the channels are desensitizing prior to opening.

The reviewer is right in mentioning that we definitely miss a sizeable fraction of the peak current for the C¹²³⁴⁵ construct (and quite probably even for C³⁴⁵). This is actually a point we already raised in the manuscript as a cautionary note (see page 9: « Of note, for constructs akin C³⁴⁵ and C¹²³⁴⁵, the fast component is so fast that we probably miss a sizeable fraction of the peak current, thereby overestimating the amplitude of the slow desensitization component and the measurement of the relative steady-state current »).

To discuss this point further (see response to point 1 from reviewer 1), we have performed simulations using either a step protocol or a ramp protocol for GABA application (with a solution exchange completed in 250 ms). This ramp protocol yields a 20-80% rise time of 23

ms on responses obtained from a simplified WT kinetic scheme, similar to the 20-25 ms range we are able to achieve in our TEVC responses.

As the reviewer points out, with a fast desensitization component in the ~ 30 ms timescale, we probably miss around half the fast desensitization component. However, the measurement of the fast desensitization kinetics remains reasonably accurate up to that stage (see new Supplementary Figure 9).

Regarding the size of currents, we now include the corresponding data in Supplementary Table 1. As the reviewer can see, the effect of mutations is rather modest regarding current amplitudes. For C¹²³⁴⁵, there might indeed be a ~ 2-3-fold reduction in the size of currents. Although we cannot be fully affirmative (the effect is not significant with a 95% confidence interval overlaying with that of the WT), this would be consistent with our experimental settings underestimating the size of peak currents due to desensitization, as illustrated in our new set of simulations (step vs ramp perfusion protocols). Of note, the peak currents for C³⁴⁵ seem rather unaffected as compared to C^{WT}.

- “.do not alter significantly the concentration-response curve of the GABA-elicited peak currents measured before the onset of desensitization. This indicates only a weak effect of the mutations on the resting-to-active state transition, and a major effect on the active-to-desensitized state transition..” Not clear how the authors say this conclusively from EC₅₀ derived from macroscopic currents. Also, wouldn't it be expected to see a left shift in dose response for gain-of-desensitizing mutants.

The putative effect of mutations on peak activation is indeed an important point to consider. However, as long as desensitization occurs on a timescale slow enough (compared to the rise time, which depends on the activation kinetics and the solution exchange time), it will not affect the measurement of peak currents. The apparent affinity (EC₅₀) from peak responses should thus not be affected by a mutation specifically altering desensitization. On the contrary, the apparent affinity measured from steady-state responses (at the plateau after prolonged GABA application), which reflects both the activation and desensitization processes, ought to be largely impacted by mutations affecting desensitization.

To illustrate this point further, we have built GABA concentration-response curves based on simulations of the WT and Mutant A using the model from the new Supplementary Figure 9. Mutant A displays a specific acceleration of desensitization which is ~ 10-fold faster than WT receptor. What can be seen in the following set of concentration-response curves is that the GABA EC₅₀ measured at the peak current is nearly unaffected in the Mutant A responses, be it in step or ramp protocols – which is expected, given that desensitization remains much slower than the 20-25 ms timescale for the 20-80% rise time of the current (peak EC₅₀ of 141 μM, 122 μM and 123 μM for WT, Mutant A with step protocol and Mutant A with ramp protocol, respectively). On the contrary, the GABA EC₅₀ measured at the steady-state response is 125-fold lower for Mutant A compared to the WT receptor (40 nM and 5 μM, respectively).

To conclude, with desensitization kinetics slow enough compared to the current rise-time, modulating desensitization alone does not affect the peak EC_{50} , while, of course, a change in the gating of the receptor (either priming or channel opening), would definitely affect the peak GABA EC_{50} .

- It is surprising that the authors use QUB to simulate currents from the various models but not use these models to directly fit the measured currents and estimate the parameters.

According to this referee suggestion, we performed a series of trial to fit automatically the experimental data. However, we found that QUB online did not yield reasonable fits of the data, even when starting with parameters that are nearby a possible solution. As an example, we departed from the following model derived from our Model 3 (i.e. with the γ and ϵ values of 100 and 10, respectively), to examine whether QUB would converge towards the numerical values for the optimized model:

We used the following constraints to fix the binding and gating rates, as well as the proportionality rules between desensitization (and recovery) rates, as stated below:

Unfortunately, QUB not only didn't converge toward a solution accounting for the experimental recording (yielding models that predict largely underestimated macroscopic desensitization kinetics), but it also completely relaxed the constraints that we wanted to enforce, yielding the following “optimized” model:

It is unclear to us why the constraining failed. But this failure prevents us from using QUB to automatically fit the experimental data.

In the present paper, we performed all the simulations by setting the parameter manually (akin Channelab, Synaptosoft, which we used for our Nat Commun 2015 paper). For the free parameters used to model the WT receptor, we want to stress that there are only 4 of them which we optimize manually (δ^+ , δ^- , δ_3^+ , δ_3^-). This number of free parameters equates the number of experimental constrains (τ_{fast} , τ_{slow} , $\%A_{fast}$, $\%I_{res}$), which are well-resolved for WT channels. We are thus confident that the modelling is actually robust, as usually accepted for quasi-linear algebraic systems. We want also to emphasize that we did not aim at getting the exact values for each individual rates, but at obtaining a reasonable fit of the whole dataset, with a set of reasonable constrains linking together the recordings of the various mutants (see below).

- What is the rationale for the initial parameters (d-, d+, d3+, d3-) used in the model. Details of how the fine tuning of the kinetic parameters done? How was the quality of the fit assessed?

The manual selection of the free parameters was performed as followed:

We simply used ballpark figures to build the initial set of parameters, already having in mind that the fast desensitization component might be mostly carried by subunits 4 and 5 (due to the C^{45} phenotype). For example, in Model II, taking into account only the two pathways linking the AO state to the AD_{45} state yields an approximate τ_{fast} of $2 \cdot (\delta^+/\delta^-) \cdot \delta^+$, while the ratio $D = \delta^+/\delta^-$ is constrained by the A_{fast} with the following approximation : $A_{fast}/I_{peak} \sim D^2/(1+2 \cdot D)$. Such approximation yields $D \sim 0.9$ and $\delta^+ \sim 0.14 \text{ s}^{-1}$.

The δ_3^+ and δ_3^- values were similarly evaluated, with δ_3^+ being set to roughly reproduce the slow component kinetics, and the ratio δ_3^+/δ_3^- being set to reproduce the residual current. We then performed iterative simulations through slight variations of the δ^+ , δ^- , δ_3^+ and δ_3^- rates to best adjust the experimental data. We assessed the quality of the fit by comparison of the simulated and experimental kinetics values τ_{fast} , τ_{slow} , $\%A_{fast}$ and I_{res} .

Overall, our model provides a good fit to the data, thereby providing a semi-quantitative interpretation of the data. While further work might result in slightly improved values for the numerical parameters, we think that the major conclusions are robust, as stated in the reply to the previous point.

The models include recovery rates from D_{fast} and D_{slow} . Recovery is not independently determined in this study, but estimated from the decay. Are there evidence in literature that the mutation does not affect recovery? The issue is for fast desensitizing constructs, the slow decay component is underestimated and hence the on and recovery rates are uncertain.

We first want to emphasize that the mutation-induced changes in recovery rates parameter are quite modest compared to the mutation-induced changes in desensitization on-rates (especially for the mutations of subunits 4 and 5: the final model does not include any effect on the recovery). Moreover, as stated in the Supplementary Discussion, we already acknowledge that our model can't be a good predictor of the receptors' recovery: it is known that recovery involves the dissociation of the agonist from desensitized receptors, while this possibility is not implemented in our model.

In addition, a few recovery kinetics were already measured in our Nat Commun 2015 paper (see Supplementary Figure 3). We showed that none of the mutations yielded a significant change in desensitization recovery. While we hadn't tested the M3-5' valine mutation, we

actually examined the effect of the M3-5' serine mutation on $\alpha 1^{N307S} \beta 2^{N303S}$ GABA_ARs, which display a weighted desensitization time constant of about 1.2 s (similar to C³⁴ in the current study), and showed no significant effect on recovery.

In the course of the current study, we wanted to have a rough idea of recovery of M3-5' valine mutant, to make sure that we waited long enough between two GABA applications to obtain full recovery.

As you can see below, for the 3 cells tested (2 C¹²³⁴⁵ cells and 1 C^{wt} cell), the recovery rates are undistinguishable ($\tau_{\text{recovery}} = 22.1, 22.7$ and 23.8 s for the two C¹²³⁴⁵ cells and the C^{wt} cell, respectively):

Given the previously published results on $\alpha 1^{N307S} \beta 2^{N303S}$ GABA_ARs, we don't think this point is critical enough to include these data in the present paper. If requested by the referee, we would of course be glad to perform additional experiments to achieve enough number of recording for incorporation of these data in a supplementary figure.

We absolutely agree that the slow component for fast desensitizing constructs is very difficult to measure reliably – and we are reluctant to draw firm conclusions from its analysis. This is all the more the case given that, as stated in the response to comment number 6 from reviewer 1, it is possible that we actually misestimate this component for several reasons (extremely small conductance of desensitized channels but not equal to zero? Potential contribution of tiny endogenous responses?). This is the reason why we don't want to push too far the models in order to fit I_{res} or τ_{slow} for strongly desensitizing mutants, as we feel this might quickly yield to a series of overstatements.

- Occurrence of multiple “Pre-sensitized open state” at same conductance level is assumed. Is there any evidence from single-channel studies, perhaps with difference in mean open times?

This is indeed an intriguing question, which we already discussed in the Supplementary discussion:

“our model makes a strong prediction when the channel transits from the open state to a more stable pre-desensitized state still capable of conducting ions. Indeed, in the open state, single-channel openings on the ms timescale are separated by brief shut times reflecting sojourns in resting or pre-active states, giving rise to a maximum open probability of about 70%-80% for $\alpha 1 \beta 2 \gamma 2$ GABA_ARs. However, once the channel enters a pre-desensitized state in our final scheme – i.e., AD3, AD4 or AD5 -, the predicted lifetime of such state should produce uninterrupted single-channel openings of several seconds for wild-type receptors. Such openings are not observed in single-channel recordings. It is likely that, when one subunit enters its desensitized conformation, the other subunits retain the ability to visit their resting conformation, thereby providing a way to maintain an open probability well below unity. Still, one could expect a strong influence of the receptor's desensitization on its single-channel gating efficacy under equilibrium conditions, e.g. in cell-attached patch-clamp recordings. It is thus striking that a recent study highlighted such phenomenon at $\alpha 1$ glycine receptors (GlyRs). Indeed, Ivica et al. showed that deletion of the intracellular domain (ICD) of homomeric human $\alpha 1$ GlyRs largely increases their efficacy of gating. At the same time, the ICD is known to affect the desensitization of $\alpha 1$ GlyRs, and the study by Ivica et al. displays single-channel clusters that seem largely lengthened by the ICD deletion, suggesting that their ICD-deleted construct shows reduced levels of desensitization (comparing the glycine and alanine recordings from panels from Figure 2B and Figure 3B in ref4). It is thus tempting to speculate that the ICD-deleted GlyRs constructs from the study by Ivica et al. show a greater stability of their pre-desensitized states with one single subunit being desensitized, resulting in both lengthened single-channel clusters and higher cluster open probability under equilibrium conditions.”

So, in a nutshell, our model potentially lacks specific transitions where a given subunit could enter its resting state (at the TMD level) while other subunits are in their D state. Also, high-quality single-channel studies are usually performed with cell-attached patch-clamp recordings, i.e. under steady-state conditions, and it is thus plausible that what is actually measured is the efficacy of gating for pre-desensitized channels.

One would need to perform single-channel recordings on outside-out patches containing only one channel, with fast perfusion of the agonist. This would enable to discriminate early gating phenotypes (with “fully active” channels) from steady-state gating phenotypes (with “pre-desensitized” channels). Such experimental approach has been used for NMDA receptors (see Erreger et al., Subunit-specific gating controls rat NR1/NR2A and NR1/NR2B NMDA channel kinetics and synaptic signalling profiles, *J Physiol* 2005 ; more recently, Amin et al., NMDA receptors require multiple pre-opening gating steps for efficient synaptic activity, *BioRxiv* 2020 doi 10.1101/2020.06.09.142687), but not for GABAA receptors (or even nAChRs / GlyRs) to our knowledge.

REVIEWERS' COMMENTS

Reviewer #1 (Remarks to the Author):

The authors have addressed my concerns.

Reviewer #2 (Remarks to the Author):

The Authors have addressed my concerns from the previous review in the main text along with a thoughtful and detailed response letter. I do not have any further concerns and recommend publication of the manuscript.

Sudha Chakrapani